# Proteogenomic analysis reveals RNA as a source for tumor-agnostic neoantigen identification

Systemic pan-tumor analyses may reveal the significance of common features implicated in cancer immunogenicity and patient survival. Here, we provide a comprehensive multi-omics data set for 32 patients across 25 tumor types for proteogenomic-based discovery of neoantigens. By using an optimized computational approach, we discover a large number of tumor-specific and tumor-associated antigens. To create a pipeline for the identification of neoantigens in our cohort, we combine DNA and RNA sequencing with MS-based immunopeptidomics of tumor specimens, followed by the assessment of their immunogenicity and an in-depth validation process. We detect a broad variety of non-canonical HLA-binding peptides in the majority of patients demonstrating partially immunogenicity. Our validation process allows for the selection of 32 potential neoantigen candidates. The majority of neoantigen candidates originates from variants identified in the RNA data set, illustrating the relevance of RNA as a still understudied source of cancer antigens. This study underlines the importance of RNA-centered variant detection for the identification of shared biomarkers and potentially relevant neoantigen candidates.

Genetic aberrations are not only centrally involved in the development of cancer but may also result in the formation of neoantigens that have the potential to mount an anti-tumor immune response. Such neoantigens can be recognized as foreign and targeted by neoantigen-specific T cells. Thus, the identification of such neoantigens is becoming increasingly important for the development of novel immunotherapies[1–5]. However, the vast majority of neoantigens are not shared between cancer patients and the validation of in silico-predicted neoantigen candidates that range in the thousands is often limited or impractical in a clinical setting. For this reason, our group reported a proteogenomic approach that combines liquid chromatography-tandem mass spectrometry (LC-MS/MS) of immunoprecipitated HLA class I (pHLA-I) peptides with whole exome sequencing (WES) of melanoma tumors for the identification and validation of such neoantigens at the protein level[6]. We were able to show that such a proteogenomic approach is feasible in fresh solid tumor material and yields a refined number of immunogenic

neoantigens. Yet the number of neoantigens that could be identified with our approach was limited and the findings had to be validated in different cancer entities.

It was reported that not only somatic mutations on coding exons represent a source of neoantigens but also non-coding transcripts, intronic regions, and splice sites[7–11]. Furthermore, RNA processing events such as RNA editing have been investigated in more detail lately. RNA editing is a widespread post-transcriptional mechanism conferring specific and reproducible nucleotide changes in selected RNA transcripts that occurs in normal cells[12] but is also involved in disease pathogenesis and is altered in cancer[13–15]. These events have been recently associated with diversifying the cancer proteome[15,16] and RNA variants derived from editing events were further investigated in more detail as a source of aberrantly expressed peptides[17,18]. As RNA regulation is mediated by *cis* regulatory elements and *trans* regulatory factors, which are often disrupted by somatic mutations or affected by oncogenic signaling[19], antigens derived from cancer-associated RNA

✉ e-mail: angela.krackhardt@tum.de

editing may represent in part true neoantigens and are therefore of high interest for targeted cancer immunotherapy. Thus, we included tumor transcriptomics in addition to WES to detect neoantigens that were derived from RNA processing events.

Furthermore, we previously showed that integrating spectral prediction features into the MS-spectra matching process during neoantigen identification, known as rescoring, is a powerful method to deal with larger search spaces and it increases the coverage and sensitivity of the analysis[20,21]. Therefore, we added the artificial intelligence algorithm Prosit and utilized a Prosit-based rescoring workflow in our pipeline for neoantigen identification[21,22].

In this work, we use a subset of 32 patients with different tumor entities that were mainly included in the previously described MASTER cohort[23] to test our improved proteogenomic pipeline in a cross-entity cohort ImmuNEO MASTER. We discover many shared DNA and RNA variants as well as tumor-associated peptides between patients independent of the tumor entity. In the majority of patients, we identify neoantigens that were predominantly derived from RNA sources. In addition, we perform T-cell phenotyping in the tumor microenvironment and show that immunogenic neoantigens correlate with increased T-cell infiltration. Thus, these data demonstrate that proteogenomic-based neoantigen identification is feasible in a cross-entity cohort and that neoantigens originating from RNA sources might represent highly relevant targets for the development of immunotherapies.

## Results

This study took advantage of a patient cohort included in the MASTER Program[23]. Detailed information about patient samples and respective analyses are described in the Methods section and are listed in Supplementary Tables 1–3.

For the identification of common tissue-agnostic immune-related hallmarks and neoantigen candidates in our cross-entity cohort ImmuNEO MASTER (Supplementary Tables 1–3 and Supplementary Fig. 1a, b), we created a general workflow for the analyses of tumor specimens which is illustrated in Fig. 1. First, tumor-infiltrating immune cells were characterized in the tumor microenvironment (TME) of fresh tumor tissue by flow cytometric immunophenotyping as well as transcriptome analyses of sorted CD8+ T cells (Fig. 1a). Next, for the respective characterization of indicated tumor specimens we used WES/whole-genome sequencing (WGS) and RNA sequencing (RNA-seq) data from patients included in the MASTER cohort or from the ImmuNEO Plus samples that were respectively analyzed at the same DKFZ facility as the samples of the MASTER cohort[23] (Fig. 1b). The analytical core of our neoantigen discovery pipeline is its proteogenomic approach. For this, we performed immunoprecipitation of pHLA-I with subsequent MS analysis for the identification of the presented immunopeptidome (Fig. 1c). We then used an optimized workflow of our previously published strategy[6] for the identification of neoantigens by combining the de novo assembled personalized genomic and transcriptomic data with the MS-based immunopeptidomic data using pFind[24] (Fig. 1d). As critical innovations we included RNA-seq data and used the artificial intelligence algorithm Prosit for increased coverage and sensitivity of our neoantigen discovery pipeline[21,22]. Immunogenicity of the identified neoantigen candidates was assessed in vitro by using patient-derived autologous or healthy donor (HD)-derived allogenic-matched T cells (Fig. 1e). Importantly, we validated the neoantigen candidates that were identified with our optimized pipeline with peptide verification and assessment of their prevalence in normal tissue expression data (Fig. 1f). Finally, in order to decipher potential clinical conditions for the identification of neoantigens which might be crucial knowledge for clinical application, we correlated the number of validated total and immunogenic neoantigens with the TME immunophenotyping data.

## The phenotype of tumor-infiltrating T cells is independent of the tumor entity

To study if we could observe tumor-agnostic immunological features in the immune TME and correlate them with clinical outcome, we performed flow cytometric immunophenotyping of fresh tumor tissues. In 17 patients, from whom enough tumor material was available, T-cell subsets were examined.

First, we looked at the relative cell numbers of CD8+ T cells per gram tumor (Fig. 2a). The two melanoma specimens and the pancreatic cancer metastasis of a patient with mismatch repair deficiency (dMMR) (ImmuNEO-11 T2) demonstrated a high amount of T-cell infiltration matching to the high mutational burden often present in these malignancies[25,26]. However, also other tumor entities, including a sarcoma specimen (ImmuNEO-5), showed high amounts of tumor-infiltrating lymphocytes (TILs) (Fig. 2a). CD8+ and CD4+ T cells predominantly consisted of effector memory T (Tem; CD45RA−CD62Llow) cells regardless of the tumor entity (Fig. 2b and Supplementary Fig. 2a, b). Moreover, the distribution of CD8+ T cell subsets and−to a lesser extent−of CD4+ T cell subsets between different metastases of a defined individual patient were highly comparable independent of their anatomical metastatic location (Fig. 2b and Supplementary Fig. 2b) and despite differences in their relative cell numbers (Fig. 2a). Since the functional state of TILs is linked to their potential anti-tumor activity, we analyzed the expression of selected activation markers (HLA-DR and CD103) and inhibitory markers (PD-1, TIM-3, and LAG-3). To account for differences in overall cell numbers and to investigate the activation status on a population level, we looked into the frequencies of activation or inhibitory markers on CD8+ and CD4+ T cells (Supplementary Fig. 2c), respectively, that express at least one marker. There was no difference in the frequencies of CD8+ T cells with activation markers between different tumor entities, and tumor specimens with high frequencies of inhibitory markers were present in carcinoma, sarcoma, and melanoma patients (Fig. 2c).

In order to identify clinically relevant transcriptional T cell signatures in our cohort, we performed RNA-seq on sorted CD8+ TILs from eight patients (Supplementary Fig. 3). Patients were grouped based on their survival data since tumor resection into a short survival (less than 1 year) and a long survival (more than 1 year) group (Supplementary Fig. 2d and Supplementary Table 1). By using gene set enrichment analyses (GSEA), we could show that pathways associated with T cell-mediated cytotoxic functions were upregulated in the long survival group, while pathways associated with general inflammatory responses were upregulated in the short survival group (Fig. 2d). In addition, to identify tissue-agnostic features that correlate with survival, the influence of each parameter on the survival of our patients since tumor resection was assessed by log-rank test and Cox's proportional hazards model (Fig. 2e and Supplementary Fig. 2e). Although the quantified numbers and frequencies of CD8+ T cells showed only a non-significant trend for a positive correlation with increased survival, the overall frequency of CD8+ T cells without inhibitory markers in the TME correlated positively with increased survival (Fig. 2e). Moreover, the frequencies of cells without activation or inhibitory markers within the CD8+ Teff subset correlated positively as well with increased survival and, consequently, a high fraction of cells with activation or inhibitory markers within this subset correlated positively with reduced survival (Fig. 2e). Of note, we observed only non-significant trends for CD4+ T cells (Supplementary Fig. 2e).

In summary, we observed that tumor-infiltrating T cells in our heterogenous pan-cancer cohort were mainly comprised of Tem cells independent of the tumor entity. Moreover, we could reproduce findings that had previously been observed in homogenous tumor cohorts, such as increased numbers of TILs in malignancies that are characterized by high mutational burden, and observed specific transcriptional pathways in CD8+ T cells that were associated with clinical outcome[27] in this cross-entity cohort.

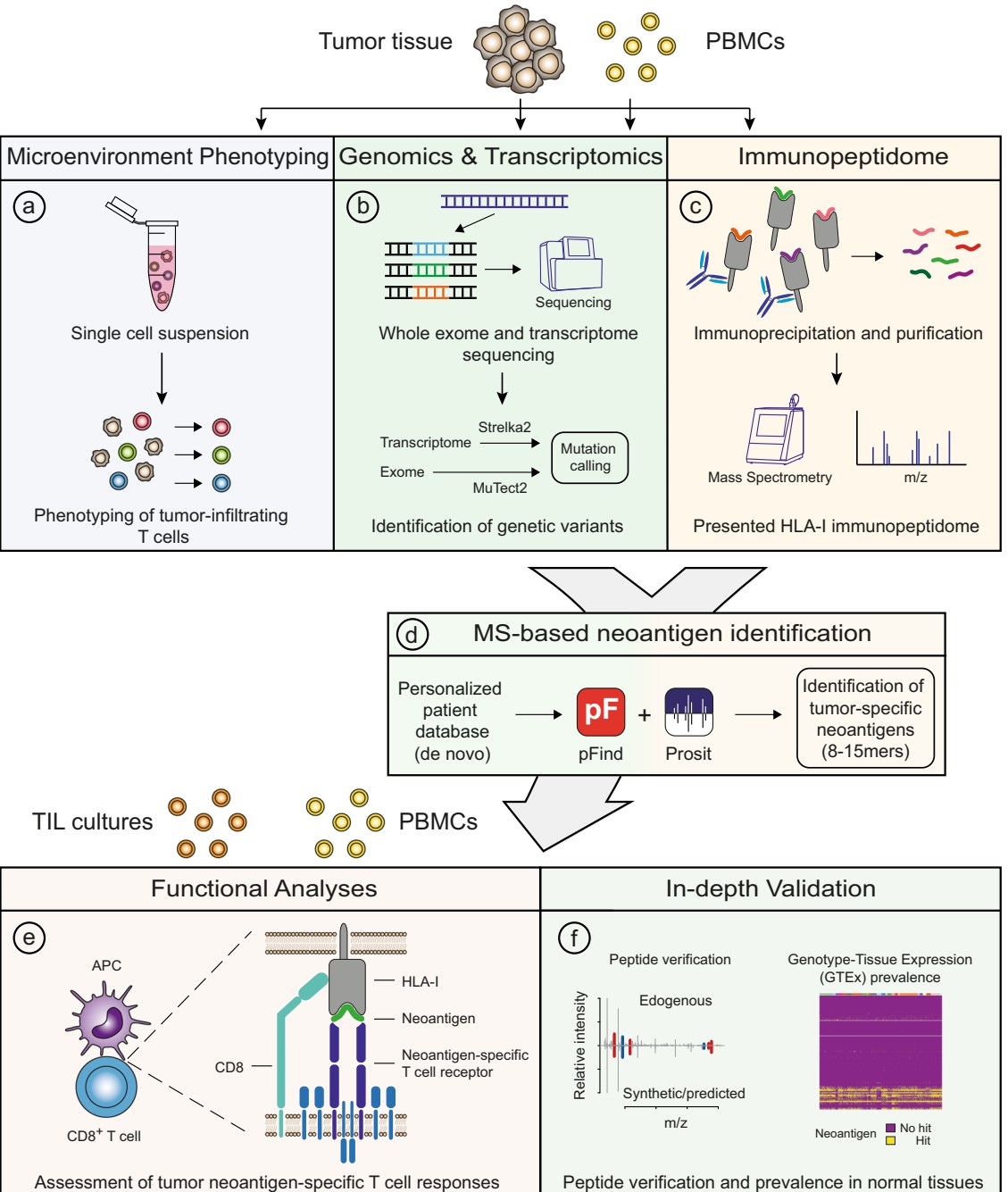

### Variants are more common at the RNA level and are often shared between different tumor entities

In the next step, we assessed the number of variants in the tumors at the DNA and RNA levels. Since these data are the basis for the identification of neoantigen candidates and will later be cross-validated by our MS-based analyses of the tumor immunopeptidomes (Fig. 1), we decided to use the data sets with unfiltered DNA and RNA variants to avoid loss of potential candidates (Supplementary Fig. 4). Of note, the majority of variants passed the filtering criteria at the RNA level for all tumor specimens but there were multiple exceptions regarding mutations at the DNA level.

The number of DNA and RNA variants varied greatly between patients but showed no clear deviation between different tumor entities in our pan-cancer cohort (Fig. 3a). On average, we

identified 302 somatic mutations per tumor, but a much higher number of variants were identified at the RNA level, with an average of 4024 variants per tumor (Fig. 3a). Of note, the majority of DNA variants were also found at the RNA level (Supplementary Fig. 5a), highlighting the power of RNA as a source for the discovery of genetic variants. In general, single-nucleotide substitutions accounted for most of the variants found at the DNA and RNA level but deletions and insertions, as well as multi-nucleotide substitutions, were also observed for some variants (Supplementary Fig. 5b). Interestingly, there was no correlation between the number of DNA and RNA variants that were identified for each tumor (Supplementary Fig. 5c), indicating that tumors with low levels of somatic mutations can still harbor a high amount of RNA variants.

**Fig. 1 | Overview of the workflow for immunophenotyping, proteogenomic, functional, and validation analyses for neoantigen identification in the cross-entity cohort.** Tumor material and peripheral blood from 32 patients included in the ImmoNEO MASTER cohort harboring diverse tumor entities was used for the following analyses: **a** Tumor microenvironment phenotyping; single cell suspensions from fresh primary tumor tissues were used for multi-color flow cytometric characterization of tumor-infiltrating T cells and FACS-sorted CD8$^+$ T cells were used for bulk transcriptome analysis (RNA-seq). **b** Genomic and transcriptomic analysis; primary tumor tissue was used for whole exome (WES)/whole-genome sequencing (WGS) and RNA-seq. Blood from the same patient served as control samples. Variants were called by MuTect2 (v4.1.0.0) from WES/WGS data and by Strelka2 (v2.9.10) from RNA-seq data and variants were filtered for single-nucleotide polymorphisms (SNPs) by using the dbSNP database. **c** Immunopeptidome analysis; fresh primary tumor tissue was used for HLA class I-bound peptide immunoprecipitation and subsequent liquid chromatography with tandem mass spectrometry (LC-MS/MS) analysis of eluted peptides. The whole HLA class I peptidome was analysed using pFind searching for 8–15mers. **d** MS-based neoantigen identification; patient-specific variant data

from (**b**) were used to generate a personalized database for matching with the MS-identified peptide sequences using pFind for the identification of neoantigen candidates. The machine learning tool Prosit was integrated in addition to rescoring the peptide spectra matching to the patient-specific personalized database. Several filtering and post-processing steps were applied for the identification of neoantigen candidates. **e** Immunogenicity assessment of neoantigen candidates; patient-derived autologous immune cells (PBMCs and TILs) and allogenic-matched healthy donor-derived PBMCs were used for immunogenicity assessment of the identified neoantigen candidates using a modified accelerated co-cultured dendritic cell (acDC) assay. **f** In-depth validation of peptides and variants; identified peptides were verified by comparison of their spectra to their synthetic peptide spectra and Prosit-predicted spectra as well as comparing their experimental and predicted retention times. RNA variants were further validated for their tumor-specificity by analysing their prevalence in normal tissue RNA-seq data obtained from the Genotype-Tissue Expression (GTEx) project[35]. APC antigen-presenting cell, FDR false discovery rate, HLA-I human leukocyte antigen class I, ORF open reading frame, m/z mass/charge number of ions, PBMC peripheral blood mononuclear cells, TIL tumor-infiltrating lymphocytes.

The higher number of variants that were detected at the RNA level compared to the DNA level could be explained in part by more non-coding sources for RNA variants, such as regulatory RNAs and pseudogenes (Supplementary Fig. 5d). However, these additional non-coding sources still did not account for this striking difference since most RNA variants were detected from protein-coding regions (Supplementary Fig. 5d). RNA editing events could present an additional source for RNA variants[12,14]. For this, we analyzed the coverage of the corresponding locus at the DNA level and nucleotide exchange patterns for all variants that were only identified at the RNA level. Indeed, for most RNA variants we could detect a corresponding canonical sequence at the DNA level (Fig. 3b), suggesting that part of these variants might be derived from RNA editing events. In fact, a considerable portion of RNA variants harbored an adenosine (A) to guanosine (G) nucleotide exchange, which has been described in the context of RNA editing events (defined by A to inosine (I) editing, where I appears as G in RNA-seq data[28])[12,15] (Fig. 3c). We observed that both DNA and RNA variants were mainly comprised of missense variants, but RNA variants consisted of more splice-site and intron variants (Fig. 3d). Although the correlation between tumor mutational burden (TMB) (DNA variants per Mb) and increased survival was not statistically significant, we observed a positive trend and the overall number of DNA variants correlated positively with increased survival in our heterogenous cohort (Supplementary Fig. 5e). There was no correlation between the number of variants that were found solely at the RNA level and overall survival (Supplementary Fig. 5e), suggesting that the sheer quantity of RNA variants does not present a prognostic biomarker for immunogenicity-associated survival.

Moreover, shared DNA and RNA variants within this pan-cancer cohort were of special interest to us as these might lead to potential common neoantigens that could be attractive targets for immunotherapy. Therefore, we investigated in how many patients each variant was detected. As expected, the vast majority of variants were found to be unique at the DNA and RNA level (Fig. 3e, f). Indeed, ~97% of variants were unique in our cohort at the DNA level (Fig. 3e) but only 89% at the RNA level (Fig. 3f). Together with the fact that we detected roughly ten times more RNA variants compared to DNA variants, this means that we could identify approximately 37 times more shared variants (detected in at least 2 patients) at the RNA level. In addition, we observed that a subset of RNA variants was shared in all patients, however, DNA variants were shared significantly less frequently and in smaller groups of patients (Fig. 3e, f and Supplementary Data 1 and 2).

To elucidate if these shared RNA variants were overlapping with each other in the same sets of patients, we focused on RNA variants that were found in at least ten tumor specimens with a minimum of two

shared RNA variants (Supplementary Fig. 5f). Overlapping shared RNA variants were not only commonly present in tumor metastases but also in different tumor entities in our pan-cancer cohort (Supplementary Fig. 5f). Although the majority of shared RNA variants in these sets were found to be exclusive, we were able to identify 59 shared variants that showed some degree of overlap. Out of these, 11 RNA variants were present in all patients and tumor metastases of our pan-cancer cohort (Supplementary Data 2, 3).

Taken together, we identified remarkably more variants at the RNA level in general and shared variants in particular, and a substantial part of additional RNA variants was likely derived from RNA editing events.

## The tumor immunopeptidomes harbor many shared cancer-associated peptides across different tumor entities

To characterize the tumor immunopeptidomes in our pan-cancer cohort, we performed immunoprecipitation of pHLA-I followed by MS analysis as previously described in ref. 6. Similar to the numbers of DNA and RNA variants, the overall numbers of peptides varied greatly between patients without a clear deviation between different tumor entities (Fig. 4a). On average, approximately 5075 peptides could be identified per tumor (Fig. 4a), with a length of 8 to 15 amino acids that were predominated by nonamers (Supplementary Fig. 6). Exemplified in four patients (ImmuNEO-4, −11, −14, −38), we analyzed the HLA anchor residues of the immunopeptides in all patients and could show that they were characteristic for the patients' HLA composition with a purity of more than 95% in the majority of patients (Supplementary Fig. 7 and Supplementary Table 4).

By focusing on peptides derived from cancer-associated genes that have been described in the Human Protein Atlas[29], we spotted that 36% of these peptides were shared between patients (Fig. 4b) and a considerable number of them were present in up to 18 patients (Fig. 4c). In addition, we analyzed peptides derived from reported cancer-testis antigens (CTAs) using the CTpedia database[30] and discovered numerous CTA peptides in our cohort (Fig. 4d). Although the majority of CTA peptides were only found to be unique in one patient, we identified multiple peptides derived from CTA-associated genes that were present in a substantial portion of patients independent of the tumor entity (e.g. *ATAD2*, *SPAG9*, *ODF2*, and *KIAA0100*) (Fig. 4d). Importantly, there was not only overlap between peptides derived from the same CTA genes across different patients, but the exact same CTA peptides could be found in multiple patients (Supplementary Fig. 8).

Investigating the immunopeptidome in this cross-entity cohort, therefore, resulted in the discovery of a number of potential tumor-associated antigen candidates for immunotherapy.

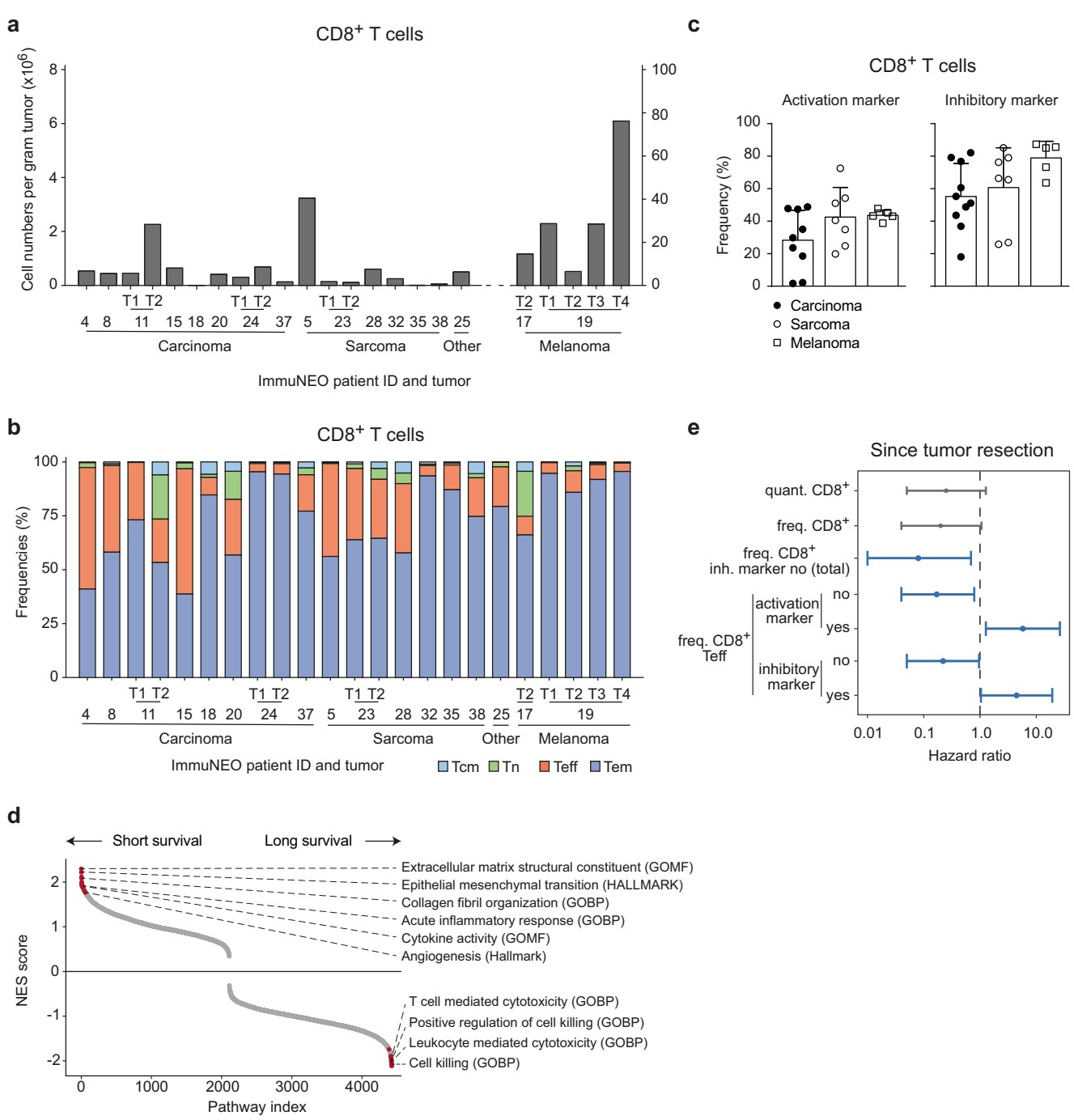

**Fig. 2 | Phenotypic and transcriptomic investigation of the immune tumor microenvironment of a defined subgroup of the ImmuNEO MASTER cohort.**
**a** Quantitative numbers of CD8+ T cells per gram tumor identified by flow cytometric assessment of fresh tumor tissue per patient grouped by tumor entity.
**b** Frequencies of different CD8+ T cell subsets of all identified tumor-infiltrating CD8+ T cells per patient grouped by tumor entity. **c** Frequencies of CD8+ T cells expressing at least one activation marker (HLA-DR, CD103) or inhibitory marker (PD-1, TIM-3, and LAG-3) for different cancer entities. Symbols depict individual tumor samples. Data were shown as mean + s.d. **d** Gene set enrichment analysis (GSEA) in the preRanked mode[83] for gene signatures differentially expressed in sorted tumor-infiltrating CD8+ T cells from bulk RNA sequencing (RNA-seq) of patients with short (below 1 year, $n = 3$) and long survival (above 1 year, $n = 5$) since tumor resection and the Hallmark and Gene Ontology gene set definitions from MsigDB v7.4[84,85]. NES scores for each pathway are depicted and significantly enriched ($p \leq 0.05$) pathways are colored in red. **e** Forest plot showing the hazard ratio

calculated by log-rank test and Cox's proportional hazards model of several phenotypic parameters for the survival of patients since tumor resection ($n = 17$). Significant correlations ($p \leq 0.05$) are highlighted in blue. For statistical analysis, only one representative tumor sample per patient was used (see core cohort Supplementary Table 1). Data were shown as hazard ratio (dot) and 95% confidence intervals (lines). **a, b** $n = 23$ tumor samples from $n = 17$ patients (see Supplementary Table 1). **c** $n = 9$ carcinoma samples from $n = 7$ patients for activation marker and $n = 10$ carcinoma samples from $n = 8$ patients for inhibitory marker; $n = 7$ sarcoma samples from six patients and $n = 5$ melanoma samples from $n = 2$ patients for activation and inhibitory marker. FDR false discovery rate, freq. frequency, GOBP Gene ontology biological function gene set, GOMF Gene ontology molecular function gene set, HALLMARK hallmark gene set, inh. inhibitory, NES normalized enrichment score, quant. quantified per gram tumor, T tumor, Tcm central memory T cells, Teff effector T cells, Tem effector memory T cells, Tn naïve T cells. Source data are provided as a Source Data file.

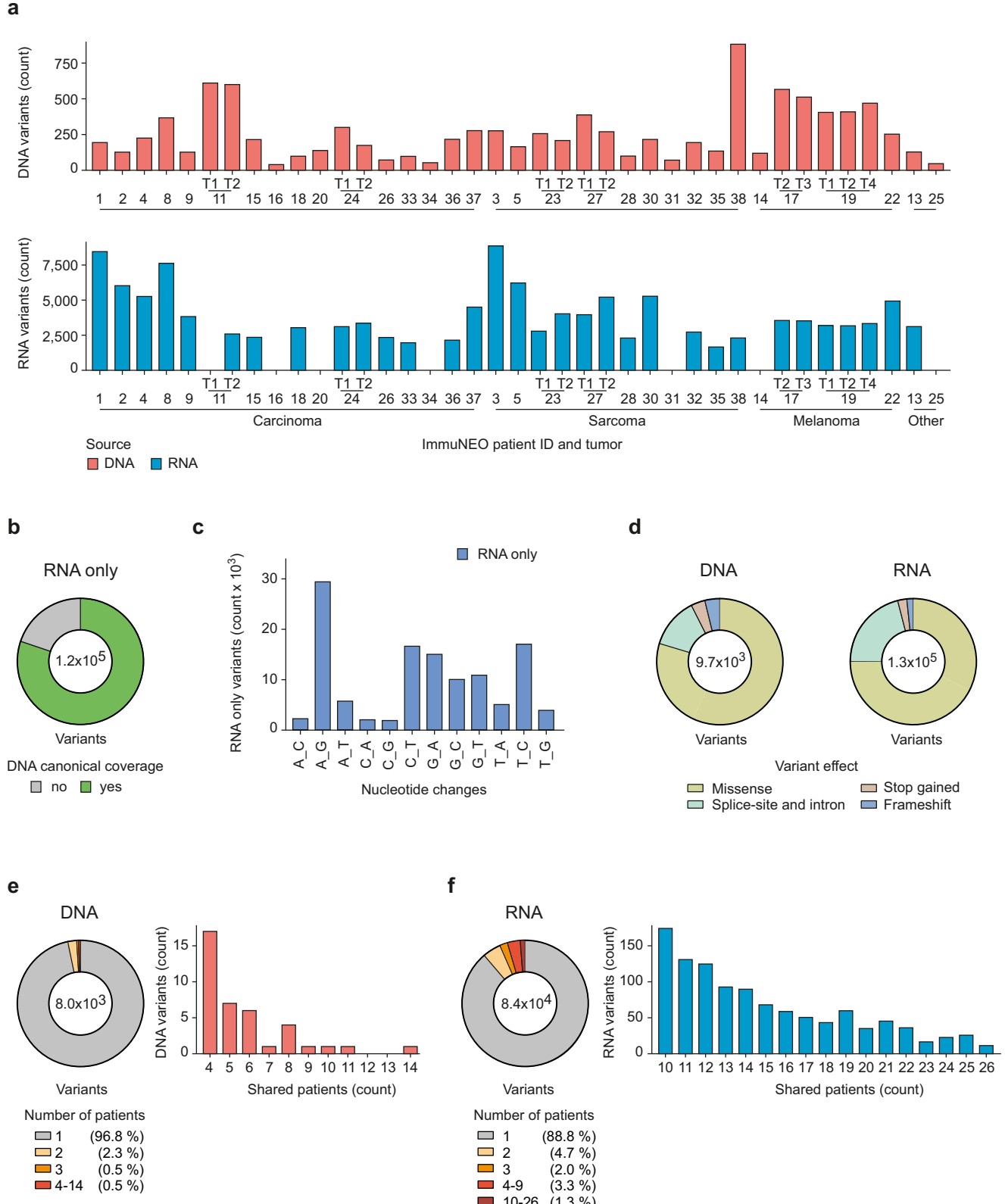

**The majority of MS-based neoantigen candidates is derived from RNA sources**

For the identification of neoantigen candidates, we have optimized our bioinformatics pipeline[6] by including additional tools such as an expanded mutation calling algorithm[31] and an improved mutation to peptide converter[32]. The peptide identification algorithm pFind[24] was used with subsequent rescoring by the machine learning algorithm

Prosit[22] (Fig. 1). Neoantigen candidates had to pass our comprehensive post-processing pipeline, which is described in detail in the method section. By utilizing a Prosit-based rescoring workflow for our proteogenomic data, we could increase the total number of identified neoantigen candidates by 13 (Fig. 5a).

With this proteogenomic pipeline, we were able to identify 90 neoantigen candidates in 24 patients across different tumor entities

**Fig. 3 | Genetic variants identified at the DNA and RNA level in tumor tissue from different cancer entities. a** Distribution of the total numbers of variants identified from DNA (upper panel) and RNA data (lower panel) identified per tumor sample grouped by tumor entity. Mutations were called by MuTect2 (v4.1.0.0) from whole exome (WES)/whole-genome sequencing (WGS) data and by Strelka2 (v2.9.10) from RNA sequencing (RNA-seq) data. SNP-filtering was performed using the dbSNP-all database. No RNA data were available for patients IN-11-T1, IN-14, IN-16, IN-20, IN-25, IN-31, and IN-34. **b** Pie chart depicting the proportion of variants only identified from RNA-seq data of all tumor samples combined where the respective canonical sequence was identified at the DNA level with coverage of ≥3 reads (green) or the respective region was not covered at the DNA level (gray, <3

reads). **c** Distribution of the nucleotide exchange pattern overall single nucleotide variants only identified from RNA-seq data of all tumor samples combined. **d** Pie charts depicting the distribution of each mutation type for variants called from all DNA (left) and RNA (right) variants. **e, f** Pie charts showing the proportions of unique and shared DNA variants (**e**) and RNA variants (**f**) between different patients. The right bar graph shows the number of variants shared by 4 to 14 patients for DNA variants (**e**) and shared by 10 to 26 patients for RNA variants (**f**) in more detail. **a–f** n = 39 tumor samples from n = 32 patients for WES/WGS data; n = 32 tumor samples from n = 26 patients for RNA-seq data (see Supplementary Table 1). T tumor. Source data are provided as a Source Data file.

(75% of all patients and 88% of patients with available RNA-seq data) with 1 to 13 identified neoantigen candidates per patient (Fig. 5b and Supplementary Data 4), highlighting that most cancer patients harbor potential targets for personalized immunotherapy. We did not observe shared neoantigen candidates between patients, however, three peptides were shared between two metastases of a melanoma patient (ImmuNEO-19) and one peptide was shared between two distinct tumor samples of a patient with dMMR (ImmuNEO-11) (Supplementary Data 4). Interestingly, we identified two neoantigen candidates in two patients (ImmuNEO-4 and −23) that were derived from shared variants in *MAP4K5* (IN_04_F, 1.5% FDR; shared between 32 tumor samples; Supplementary Data 2) and in *AC024075.2* (IN_23_A, 4.3% FDR, shared between 24 tumor samples; Supplementary Data 2), respectively. Since both of these shared variants were able to yield a pHLA-I that was presented in at least one patient, it is possible that these two peptides are presented in other patients with the variants but were missed due to detection limitations of the patients´ immunopeptidomes.

The peptide length of all identified neoantigen candidates ranged from 8 to 14 amino acids with nonamers predominating (Fig. 5c). Perhaps most strikingly, out of 90 identified neoantigen candidates 79 were derived exclusively from RNA variants, while only three originated exclusively from DNA variants, and eight were shared between both sources (Fig. 5d). Comparable to the overall number of RNA only variants, we could detect a corresponding canonical sequence at the DNA level for the majority of identified neoantigen candidates that were derived exclusively from RNA variants (Fig. 5e). Moreover, many of these variants also harbored an A to G nucleotide exchange pattern that has been associated with RNA editing (Fig. 5f). This suggests that RNA-altering mechanisms (e.g., RNA editing) could be an important source for the formation of neoantigens. Regarding the variant effect of the variants that gave rise to the neoantigen candidates, missense variants were still most abundant, however, splice-site and intron variants were more prevalent compared to overall detected variants (Fig. 5g, left). The majority of neoantigen candidates were derived from protein-coding regions but a substantial amount was also derived from non-coding regions such as pseudogenes and lncRNAs (Fig. 5g, right).

Taken together, our data indicate that MS-based identification of neoantigen candidates is feasible in the majority of cancer patients with tumor RNA representing an important source for the detection of peptide ligands derived from variants.

### Identified neoantigens derived from RNA sources are immunogenic in a set of patients independent of the tumor entity
To assess the immunogenicity of the identified neoantigen candidates, we evaluated T-cell responses against 78 neoantigen candidates from 21 patients in an in vitro assay with autologous or allogenic HLA-matched peripheral blood mononuclear cells (PBMCs) or expanded TILs by ELISpot analysis (Supplementary Fig. 9a).

Out of 78 examined neoantigen candidates, 21 were capable of inducing T cell responses (27% of all tested neoantigen candidates) in either an autologous PBMC (Fig. 6a, left), expanded TIL (Fig. 6a, right), or an allogenic-matched PBMC (Fig. 6b) culture setting (Fig. 6c and

Supplementary Data 4). The majority of immunogenic neoantigens were identified by using autologous PBMCs and only three immunogenic neoantigens could be identified with expanded TILs (Fig. 6a). This highlights the difficulties known for TIL cultures that could be explained by either insufficient expansion or a dysregulated and exhausted T cell phenotype of the expanded TILs, thus, preventing a proper T cell response against the presented neoantigen candidates. Although allogenic-matched PBMC cultures are challenging, especially with respect to donor selection, we tested a small set of neoantigen candidates (n = 10) and could confirm the immunogenicity for four neoantigens that were immunogenic in the autologous setting and even identified one additional immunogenic neoantigen (candidate 19A) (Fig. 6b). Of note, there was no enrichment observed regarding the frequency of immunogenic neoantigens out of the pool of neoantigen candidates that were identified by either of the two processing workflows or by both of them (Fig. 5a and Supplementary Data 4).

Importantly, all 21 immunogenic neoantigens were identified from RNA sources, with 20 detected exclusively from RNA variants and only one from both RNA and DNA variants (Fig. 6d). In line with our findings for RNA-only variants and neoantigen candidates, we observed that the majority of immunogenic neoantigens harbored a detectable canonical sequence at the DNA level (Supplementary Fig. 9b) and a substantial portion had an A to G nucleotide exchange pattern (Supplementary Fig. 9c). This supports our hypothesis that RNA-altering mechanisms might be implicated in the formation of neoantigens that are capable of inducing T cell responses in patients. Moreover, the variant effect and the transcript type of the variants that gave rise to the immunogenic neoantigens were highly comparable to the distribution of neoantigen candidates as well (Fig. 6e). Overall, we observed immunogenicity of neoantigens regardless of the patient's tumor entity, including patients with carcinoma, sarcoma, and melanoma (Supplementary Fig. 9d and Supplementary Data 4), indicating that the identification of immunogenic neoantigens is not limited to specific tumor entities.

In summary, we identified immunogenic neoantigens in a quarter of all patients of our pan-cancer cohort independent of the tumor entity by using a proteogenomic pipeline that utilizes RNA transcriptomics of tumor specimens for variant identification.

### In-depth validation allows for fine-tuned selection of highly promising neoantigen candidates
To increase the likelihood of detection of potential neoantigens with our proteogenomic pipeline, we used relaxed criteria preliminary as is the common practice in the field due to the very low prevalence of these targets. However, this may increase the risk of discovering false neoantigen candidates. We, therefore, used our proteogenomic pipeline as a hypothesis generator for the identification of candidates requiring further validation. Our validation strategy applied here could serve as potential guidance for the prioritization of neoantigen candidates with respect to clinical translation.

For further validation, we focused on both proteomic and transcriptomic data sets. In order to further confirm the spectra, we

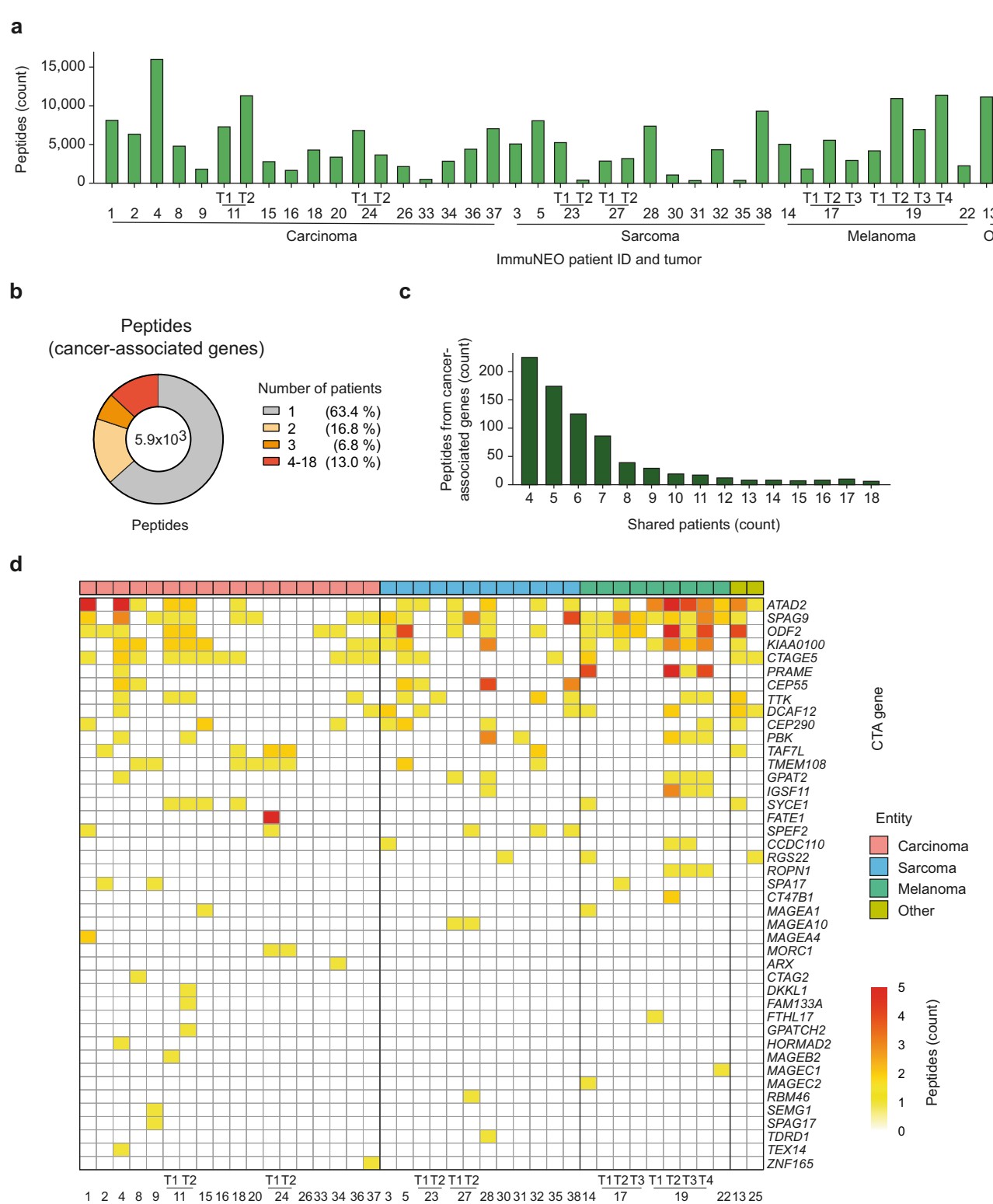

**Fig. 4 | Analysis of the HLA class I tumor immunopeptidomes. a** Distribution of the total number of unique HLA class I peptides identified per tumor sample grouped by tumor entity. Peptides bound to HLA class I molecules on the surface of tumor cells were isolated by immunoprecipitation and sequenced by liquid chromatography with tandem mass spectrometry (LC-MS/MS). Peptide sequences were then mapped with 1% FDR to the Ensemble92 protein database using pFind (v3.1.5) and unique sequences have been filtered. **b** Pie chart showing the proportion of unique and shared peptides originating from cancer-associated genes (ProteinAtlas) between patients. **c** Bar graph depicting the number of peptides shared by 4 to 18 patients in more detail. **d** Heatmap depicting the numbers of unique peptides found per cancer-testis antigen (CTA) gene in each tumor sample. Genes were sorted by the total number of peptides identified overall patients and samples were grouped by entity. **a**–**d** n = 41 tumor samples from n = 32 patients (see Supplementary Table 1). FDR false discovery rate, HLA human leukocyte antigen, T tumor. Source data are provided as a Source Data file.

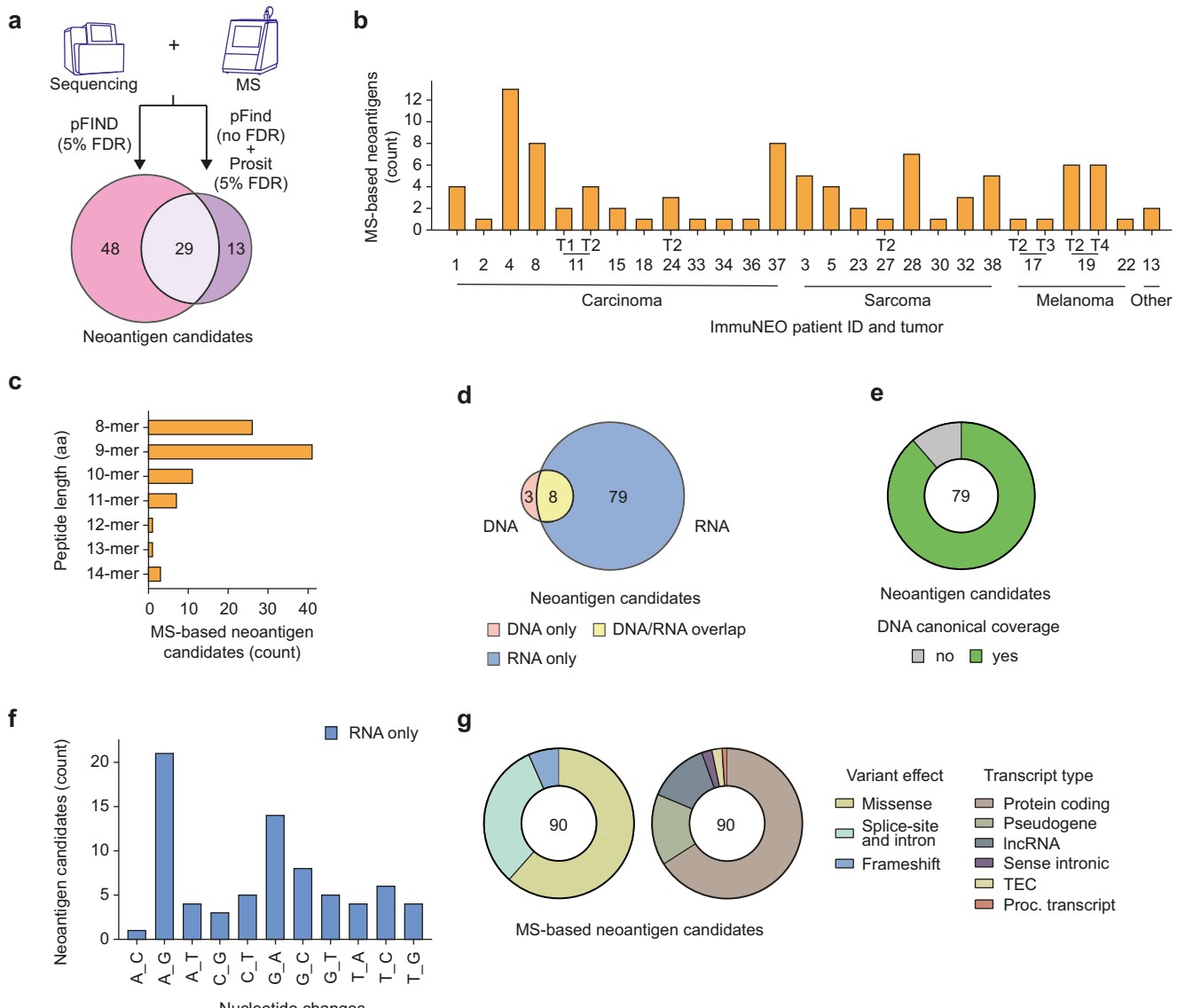

**Fig. 5 | Proteogenomic identification of neoantigen candidates. a, b** Number of identified neoantigen candidates based on the bioinformatics tool that they were identified with (**a**) and per tumor sample and grouped by tumor entity (**b**). pFind (v3.1.5)[24] was used at 5% FDR on spectral level for the identification of non-canonical 8–15mer neoantigen candidates. The machine learning tool Prosit[22] was integrated in addition to rescoring the peptide spectra matching to the patient-specific personalized database using unfiltered pFind data as input. $n = 39$ tumor samples from $n = 32$ patients were analysed in total; $n = 27$ tumor samples from $n = 24$ patients harbored $n = 90$ neoantigen candidates. **c** Bar graph showing the length distribution of all identified neoantigen candidates in amino acids (aa). **d** Source (DNA or RNA data) of the variants that the identified neoantigen candidates were derived from. **e** Pie chart depicting the proportion of neoantigen candidates identified only from RNA sequencing (RNA-seq) data where the respective canonical sequence was identified at the DNA level with coverage of ≥3 reads (green) or the respective region was not covered at the DNA level (gray, <3 reads). **f** Distribution of the nucleotide exchange pattern of all variants that yield neoantigen candidates identified only from RNA-seq data. **g** Distribution of each mutation type (left) and biotype (right) of all variants that yield neoantigen candidates. **a–g** $n = 39$ tumor samples from $n = 32$ patients were analysed in total; $n = 27$ tumor samples from $n = 24$ patients harbored $n = 90$ neoantigen candidates; $n = 3$ neoantigen candidates from DNA variants; $n = 8$ neoantigen candidates from DNA and RNA variants; $n = 79$ neoantigen candidates from RNA variants. aa amino acids, MS mass spectrometry, Proc. processed, T tumor, TEC to be experimentally confirmed. Source data are provided as a Source Data file.

performed peptide sequence verification by comparing the endogenous MS spectra of the neoantigen candidates from the tumor with the MS spectra of their cognate synthetic peptides. In addition, we performed the same comparison using Prosit-predicted fragment ion intensities. We defined neoantigen candidates with a normalized spectral contrast angle (SA)[33] of at least 0.7 with the synthetic or Prosit-predicted spectra as verified according to a previous study by ref. [34]. Out of 88 tested peptides, we found that 41 could be verified using these criteria (Fig. 7 and Supplementary Data 4, 5). 19 candidates were close to the SA cutoff and may still represent true peptide-spectrum matches although further confirmation might be necessary. Others

with low SA values may not be safe enough for therapeutic targeting as the spectra may have been mistaken and could actually correspond to different peptides. The candidates that could not be verified are likely a result of our relaxed criteria. However, while stricter criteria with an FDR below 1% for peptide spectrum matching resulted in less peptides with a SA value below the 0.7 cutoff, we still observed many peptides above the SA cutoff with q-values between 1 and 5% (Supplementary Fig. 10, red rectangle) that would have been missed if we would have used stricter criteria primarily rather than subsequent peptide verification. Moreover, we compared the LC experimental retention times (RT) with the predicted RTs and use this as an additional scoring

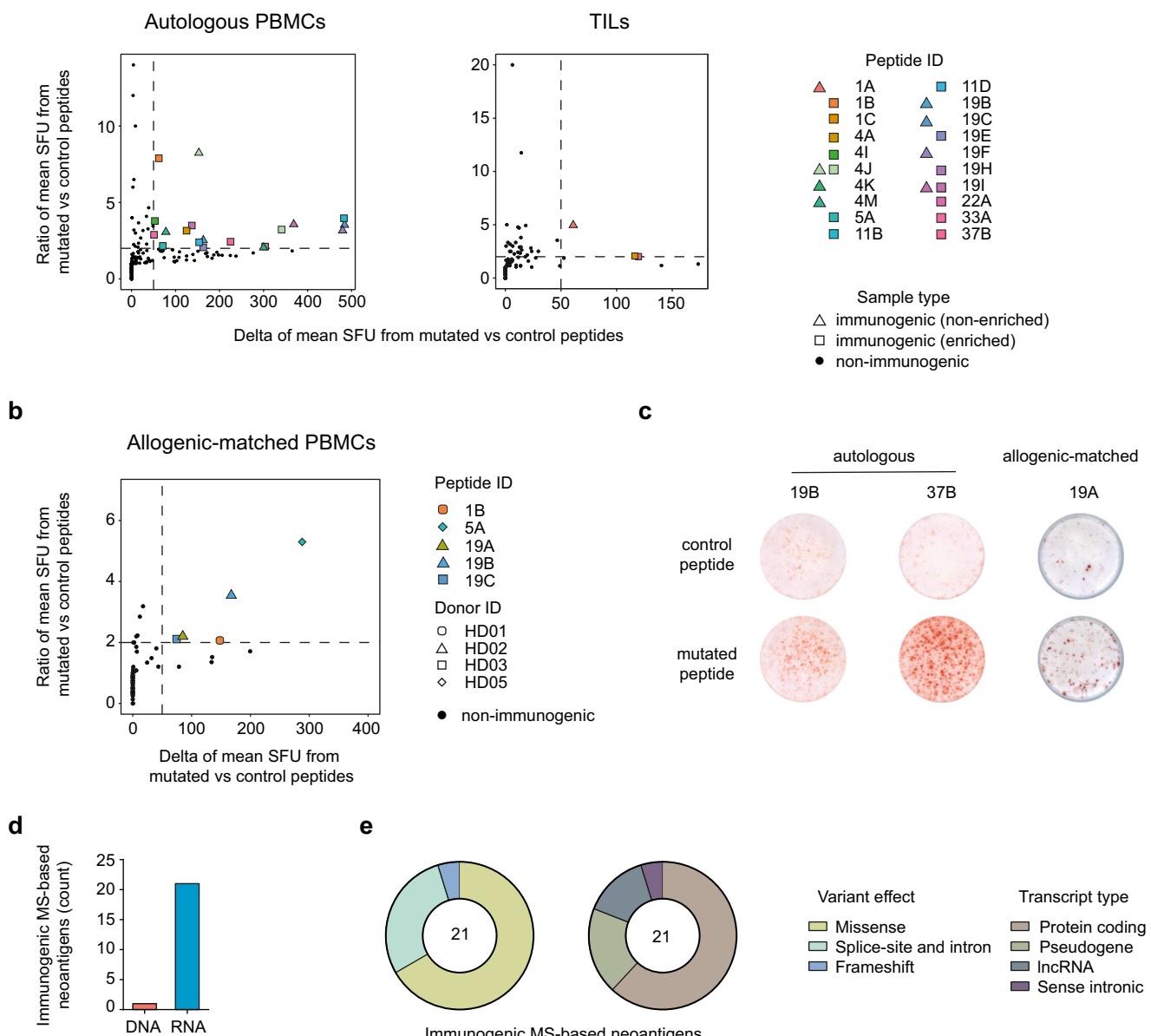

**Fig. 6 | Immunogenicity assessment of neoantigen candidates. a, b** Summary of immunogenicity assessment data from all performed modified accelerated co-cultured dendritic cell (acDC) assays for neoantigen candidates by ELIspot analysis using patient-derived PBMC (left plot) or TILs (right plot) (**a**) and allogenic-matched healthy donor PBMCs (non-enriched) (**b**). Mean IFN-γ spot forming units (SFU) for T cells tested against the mutated peptide (test condition) and tested against a control peptide (control condition) were calculated and the ratio as well as the difference of the mean SFU have been determined. Values are shown for every peptide and PBMC or TIL aliquot tested. Highlighted are peptides that elicit an immune response where the ratio of SFU is >2 and the difference of SFU is >50. Autologous LCLs or allogenic HLA-matched cells (LCLs or HLA-transduced cell lines) were used as target cells. Negative values (when controls show more spots than the test condition) were set to 0 for better readability. **c** Representative IFN-γ ELIspot data showing spots per well for autologous and allogenic-matched PBMCs tested against a control peptide (top) and the indicated neoantigen candidate (bottom). **d** Genetic origin (DNA or RNA data) of the variants that the identified immunogenic neoantigens were derived from. **e** Distribution of each mutation type (left) and biotype (right) of all variants that yield immunogenic neoantigens. **a, d, e** $n = 78$ neoantigen candidates from $n = 24$ patients were analysed in total; $n = 8$ patients harbored $n = 20$ immunogenic neoantigens; $n = 17$ immunogenic neoantigen candidates from autologous PBMC cultures; $n = 3$ immunogenic neoantigen candidates from TIL cultures; $n = 23$ tumor samples from $n = 17$ patients for immunophenotyping data. **b** $n = 10$ neoantigen candidates from $n = 4$ patients were analysed in total; $n = 5$ immunogenic neoantigen candidates from allogenic-matched PBMC cultures. MS mass spectrometry, PBMCs peripheral blood mononuclear cells, SFU spot forming units, TIL tumor-infiltration lymphocytes. Source data are provided as a Source Data file.

parameter for peptide verification (Fig. 7 and Supplementary Fig. 11). Half of the candidates' experimental RT matched with the predicted RT ($n = 45$ candidates) and there was a portion of candidates that fell into an RT range below 20 min where the prediction algorithm was lacking accuracy based on all measured peptides ($n = 17$). Candidates with RT

deviation therefore may not necessarily be excluded if they pass the SA cutoff.

Since RNA editing is a physiological process that plays an important role in health, we applied a further neoantigen confirmation step by analyzing the prevalence of our candidate variants in normal tissues

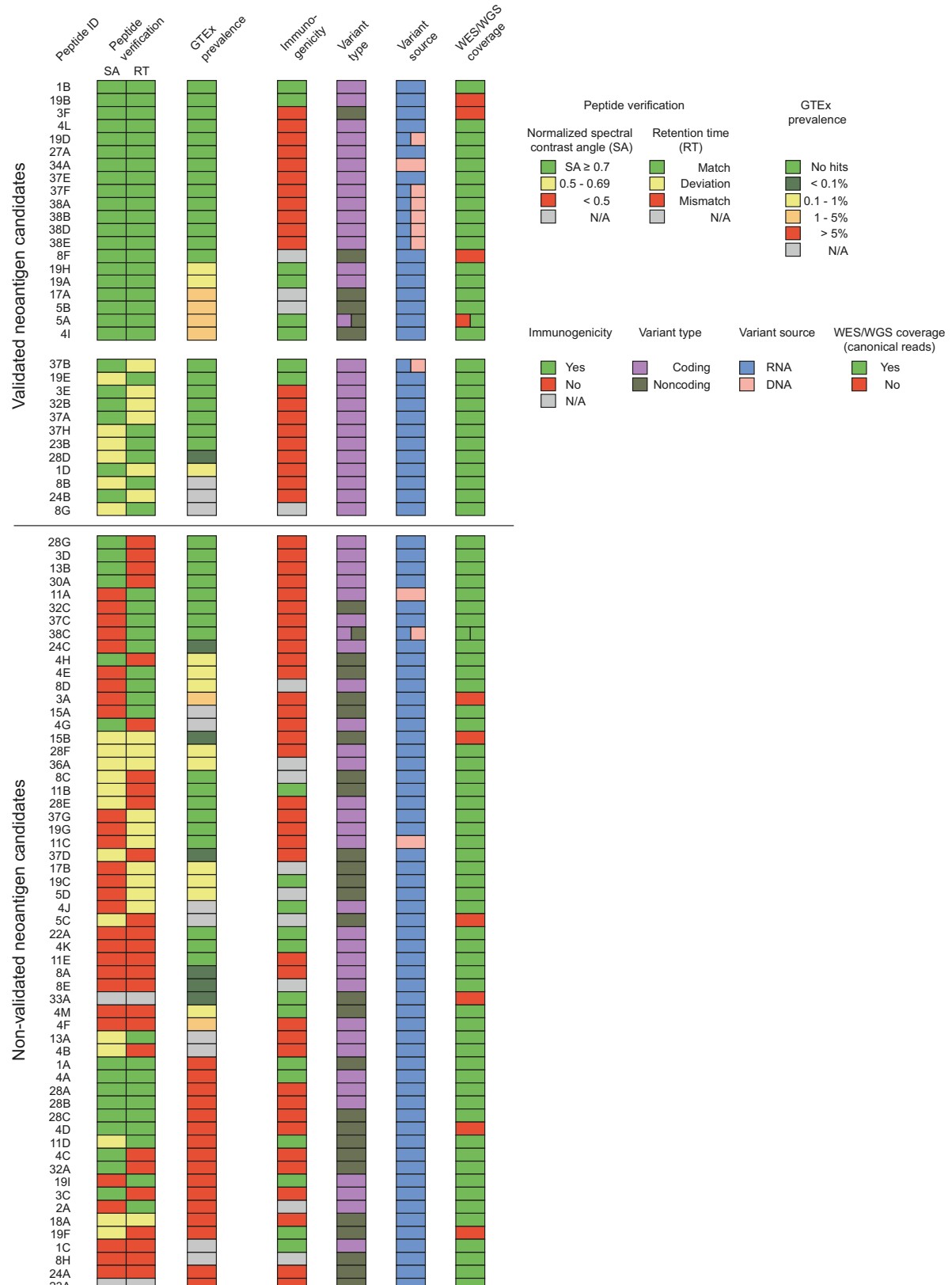

using public databases. Over 10,000 RNA-seq samples from 30 different tissues obtained from the Genotype-Tissue Expression (GTEx) project[35] were analyzed for the presence of the 90 neoantigen variants. Out of 90 candidates, 38 were completely absent in normal tissue RNA-seq samples in our extensive GTEx analysis and might represent highly interesting neoantigen candidates based on this criterion. The other 52

candidates either showed a high prevalence (n = 16 candidates; found in more than 5% of samples), intermediate prevalence (n = 6 candidates; found in 1 to 5% of samples), low prevalence (n = 12 candidates, found in 0.1 to less than 1% of samples), very low prevalence (n = 7 candidates; found in less than 0.1% of samples) in normal tissues or were defined as not available (N/A) based on expression data (n = 9

**Fig. 7 | In-depth validation of neoantigen candidates.** Validation of all 90 neoantigen candidates based on peptide verification and neoantigen candidate variant prevalence in normal tissues. Peptides were verified by comparison of their spectra to their synthetic peptide spectra and Prosit-predicted spectra. The best normalized spectral contrast angle (SA) of both methods was used and grouped into peptide-spectrum matches (green, $n = 41$), potential matches (yellow, $n = 19$), and mismatches (red, $n = 28$). Retention times (RT) of each peptide were predicted using Prosit and compared to the experimental RTs as an additional scoring criterion for peptide verification. Based on the error distribution for all peptides (see Supplementary Fig. 11) $n = 45$ peptides were considered matching (green). For $n = 17$ peptides no accurate RT error could be predicted according to the distribution of canonical peptides and these were considered deviations (yellow). The prevalence of all neoantigen candidate variants in normal tissues was assessed in RNA expression data obtained from the Genotype-Tissue Expression (GTEx) project[35] ($n = 10,269$ samples from 30 different normal tissues). Candidate variants were either absent (green, $n = 38$), showed very low prevalence (dark green, $n = 7$), low prevalence (yellow, $n = 12$), intermediate prevalence (orange, $n = 6$), a high prevalence ($n = 16$) in normal tissues or were defined as not available (N/A) based on expression data ($n = 9$ candidates; where the variant locus is expressed in less than 5% of normal tissue samples with at least three reads). Validated neoantigen candidates were comprised of promising candidates (top, $n = 20$) and potentially promising candidates that need further verification of their peptide sequence or prevalence in normal tissues (middle, $n = 12$) (in total $n = 32$), and non-validated neoantigen candidates that either lacked sufficient peptide verification or showed high prevalence in normal tissues (bottom, $n = 58$) were separated by a line. The immunogenicity (Fig. 6) of each neoantigen candidate, the variant type, the source of variant identification, and the coverage at the DNA level (at least three canonical reads) are displayed. RT retention time, SA spectral contrast angle, WES whole exome sequencing; WGS whole-genome sequencing. Source data are provided in Supplementary Data 4.

candidates; where the variant locus is expressed in less than 5% of normal tissue samples with at least 3 reads) (Fig. 7 and Supplementary Fig. 12a). Of note, DNA-derived variants were not detected in normal tissues with our comprehensive analysis as expected (Fig. 7). To screen for rare patient-specific variants that might have been missed by our GTEx analysis, we also investigated the total RNA-seq data from CD8+ TILs that we used in our GSEA analysis (Fig. 2d) for the presence of the neoantigen candidate variants. In this subset of patients, 8 neoantigen candidate variants were detected in the CD8+ TILs but only one of them (candidate 19F) was found in the patient it was originally identified in the tumor (Supplementary Fig. 12b), meaning both the RNA variant and the immunopeptide were detected with our pipeline in this patient. Importantly, this neoantigen candidate also showed high prevalence in our GTEx analysis and did not pass the validation process. Two neoantigen candidate variants (candidate 4B and 13 A) were detected in our total RNA-seq analysis that were not found in our GTEx analysis, likely due to low expression of the variant locus (Supplementary Fig. 12b). In addition, we investigated tumor-associated RNA-overediting by comparing the variant frequency for the neoantigen candidates in tumor and normal tissues. Selected candidates (2A, 19A, 19F, 28B) may represent potential tumor-associated RNA-overediting but the majority of candidates' variant frequency was not elevated compared to normal tissues (Supplementary Fig. 13).

By incorporating these quality controls for peptide verification and prevalence in normal tissues into our proteogenomic pipeline for neoantigen identification, we grouped our neoantigen candidates into 20 highly promising candidates (Fig. 7, top), 12 potentially promising candidates that need further verification of their peptide sequence or prevalence in normal tissues (Fig. 7, middle), and 58 candidates that either lack sufficient peptide verification or are commonly found in normal tissues (Fig. 7, bottom). Most of the 32 validated candidates were identified by both pFind and Prosit and were predominantly nonamers (Supplementary Fig. 14a–c). Importantly, the majority of validated neoantigen candidates were derived from RNA variants that lacked somatic mutations and could constitute RNA-editing events (Supplementary Fig. 14d–f). Protein-coding variants were by far the most common transcript type among the validated candidates (Supplementary Fig. 14g). When looking at binding predictions for our identified neoantigen candidates with NetMHC 4.0[36] and MHCFlurry[37] (Supplementary Data 4), 90% of highly promising candidates (18 out of 20), 58% of potentially promising candidates (7 out of 12), and 51% of non-validated candidates (30 out of 59) were predicted with at least one algorithm as binders (percentile rank <2% or predicted binding affinity <500 nM). This observed enrichment of predicted binders for validated candidates supports a good performance of our peptide verification approach. Out of the 32 validated candidates, 8 elicited an immune response of CD8+ T cells. Finally, in an effort to link the level of identified neoantigens (Figs. 5–7) with the immune activity in the TME and the level of detected immunopeptides of our patients, we

performed a Spearman's rank correlation test with our immunophenotyping (Fig. 2) and immunopeptidomic data (Fig. 4). Since all neoantigen candidates were matched to the presence of pHLA-I mass spectra, both the number of neoantigen candidates and immunogenic neoantigens correlated strongly with the size of the detected immunopeptidome (Supplementary Fig. 15). The overall number of non-validated neoantigen candidates also correlated slightly with the total frequency of CD3+ T cells and CD8+ Teff cells in the TME (Supplementary Fig. 15). Importantly, the number of immunogenic validated neoantigen candidates did not only correlate stronger with the total frequency of CD3+ T cells and CD8+ Teff cells in the TME, but we also observed a strong correlation with CD8+ T cells as well as a generally more exhausted phenotype (Supplementary Fig. 15). Thus, immunogenic validated neoantigen candidates correlated with a more immune-active TME with high T-cell infiltration in our cohort.

In summary, we added comprehensive validation analyses to our neoantigen identification pipeline that provide important parameters for the selection of targets for immunotherapy. These immunogenic altered peptides correlated with T-cell infiltration and potentially an exhausted T-cell phenotype.

## Discussion

The clinical application of personalized cancer immunotherapies based on neoantigens is benefitting greatly from the recent advances in mRNA-based vaccines[4] and cellular immunotherapy[38]. However, the identification of tumor-specific and therapy-relevant targets is still critical. This is an area of research that mainly focused on cancer genomics and bioinformatics epitope prediction models for the identification of potential neoantigens in the past[1] but might benefit greatly from combinatorial approaches like proteogenomics that have been applied by other groups[7,10] and us[6,21]. In this study, we showed that RNA is an important source for the identification of neoantigens and shared tumor antigens with our improved proteogenomic pipeline in an extensively characterized pan-cancer cohort. By combining proteogenomics with phenotypic and functional analyses, we linked the identified candidates to immunological features and assessed their potential to induce T cell-driven immune responses. Moreover, we added validation analyses to our proteogenomic pipeline that might guide the selection process of promising neoantigen candidates for downstream preclinical testing.

Despite the relatively small size of this cohort and the high diversity with respect to tumor entity, disease stage, treatment history, age, and gender, we were able to confirm biomarkers with prognostic significance which have been already established for a number of distinct malignancies, indicating that these biomarkers have a strong prognostic power. When looking at the TMB as a prognostic biomarker, we could confirm a significant positive correlation between the number of somatic mutations and patients' survival, as previously shown for several different cancer entities as well as selected cross-

entity studies[39–42]. In addition, we observed that high levels of CD8+ T cells expressing inhibitory markers, previously shown as an indication for a dysfunctional T cell state in the TME[43], correlated with poor clinical outcomes.

To increase the number of identified neoantigens from our previously published proteogenomic strategy[6], we integrated tumor RNA as an additional source for variant detection. Including RNA-seq in our pipeline has two advantages. First, RNA-seq has been shown to complement WES in calling somatic mutations in glioblastoma multiforme to broaden the scope of discoveries[44]. Second, RNA-seq is able to detect variants that are not occurring at the DNA level but are derived from RNA processing events like alternative splicing and RNA editing[45,46]. It has been previously reported that RNA editing events and RNA dysregulation lead to the diversification of the cancer proteome[15,16] and in fact, we substantially increased the number of variants and neoantigens by including RNA-seq in our pipeline. Variant detection using RNA-seq is already utilized in a number of studies for the identification of neoepitopes[7,17] but comes with its own limitations, in particular for variants derived from RNA processing events since they cannot be validated by matched-normal DNA samples. In addition, obtaining matched-normal RNA samples from the same tissue as the tumor is similarly limited as it might be either not available or may be influenced by the tumor activity and transcriptional profile of the surrounding tissue. To exclude false positive RNA variants based on single-nucleotide polymorphisms (SNPs), we used a methodology of combining tumor RNA-seq with normal WES data that has been shown to be most effective for calling RNA variants[47]. We thereby excluded frequent population SNPs. Since that still did not control for false positive RNA variants from RNA processing events, we overcame this limitation by matching the RNA variants to the MS spectra from the tumor pHLA-I and thereby performed cross-validation of the neoantigen candidates. In addition, we performed peptide verification of the neoantigen candidates and analyzed over 10,000 RNA-seq samples from normal tissues as well as total RNA-seq samples from CD8+ TILs in a subset of our patients for the prevalence of the neoantigen candidate variants. Of note, due to this subsequent validation, less stringent mutation calling algorithms for RNA but also DNA variant detection and peptide spectra matching were used that increase the search space for potential neoantigen identification. Therefore, false positive hits may have been still not completely excluded here. However, using our sensitive algorithms for the detection of DNA and RNA variants, we were able to identify variants that occurred not only in individual patients but were shared in a substantial number of patients at the DNA and especially at the RNA level. Although variants at the RNA level were less personalized compared to somatic mutations, we did not observe shared RNA-derived neoantigen candidates. Moreover, some of the RNA-derived candidate variants were highly prevalent in normal tissues, indicating that a portion of shared RNA variants could also stem from physiological RNA editing at this stage. Therefore, the origin of these shared RNA variants needs to be further elucidated to assess their potential as common targets for immunotherapy in the future.

The strength of our neoantigen discovery platform, the matching of MS-spectra to variants, is also its bottleneck because the number of identified neoantigens strongly correlated with the size of the immunopeptidome. Therefore, improving MS-based neoantigen detection is paramount and there are three avenues that can be addressed. (1) Optimizing artificial intelligence tools for the matching and rescoring of MS spectra (like Prosit) will enhance their potential for neoantigen discovery. (2) Improving protocols for sample preparation and immunoprecipitation of pHLA-I might result in a higher yield of detected peptides. (3) Increasing the sensitivity of MS instruments will likely have the biggest impact in the future[48].

The number of neoantigen candidates that we identified was small compared to the thousands of hits that were reported with epitope prediction models[1,2]. However, in our study, ~29% of the tested validated candidates elicited a T-cell response in vitro, a far greater number than could be expected from any epitope prediction approach. Thus, drastically reducing the need for large-scale immunogenicity testing that would not be feasible in a clinical environment. Importantly, the assessment of immunogenicity does not serve as validation of the neoantigen candidates. More than half of the immunogenic candidates did not pass our validation criteria either at the peptide verification level—and might constitute a T cell response of the vast TCR repertoire against a random foreign peptide—or they were highly prevalent in normal tissues and could represent autoreactive epitopes. Autoreactive epitopes could occur in cancer patients as a result of excessive apoptosis or treatment, and increased RNA editing has been reported as a source for autoantigens in systemic lupus erythematosus[13], which might also play a role in cancer. Going forward, we suggest to implement peptide verification and normal tissue expression data analysis prior to immunogenicity assessment. Although using large repositories with normal tissue expression data like GTEx[35] is a very powerful approach, we propose to combine this with total RNA-seq analysis of liquid biopsies from the patients since most RNA-seq data from GTEx is mRNA-based and does not cover intronic regions. Therefore, coverage of the variant locus needs to be confirmed if using these repositories for the validation of neoantigen candidate variants as we have done for our analysis. Personalized tissue-specific editing events that are not present in the blood could still not be excluded with this approach but the overall risk for on-target, off-tumor toxicity will be lowered.

Correlations with the immunophenotyping data from the TME indicate that the presence of immunogenic validated neoantigen candidates might be associated with increased tumor T-cell infiltration and activity. This could make these immunogenic candidates particularly interesting but our sample size here is small and this needs to be corroborated in further studies. Moreover, immunogenicity testing in autologous T cell assays has the inherent risk of a lack of an immune response to the presented peptide because of T cell dysfunction[49], suggesting that some neoantigen candidates that did not induce a T cell response might actually be potentially immunogenic. More sensitive assays are therefore necessary and combined single-cell RNA and T-cell receptor (TCR) sequencing shows great promise for this need[49,50].

Neoantigen candidates derived from RNA variants have been previously reported[7–11,17,18,51] and may represent missing targets in studies where suspected neoantigens could not be detected by focusing only on WES[50]. Indeed, the majority of neoantigens in our cohort were derived only from RNA variants and we observed a high number of A-to-I (detected as G in sequencing) modifications typical for RNA editing events[17,28]. A-to-I editing catalyzed by adenosine deaminases acting on RNA (ADARs) is the most common form of RNA editing and clinically relevant A-to-I editing events have been reported in different cancer entities in particular in non-coding regions[14]. Interestingly, the majority of RNA-derived neoantigen candidates in our study were discovered from coding regions. This might be due to our cross-examination with the detected cognate HLA-peptide ligands in our approach that factors in not only the expression but also the translation of each variant. Moreover, also other editing patterns such as cytidine to uridine (C-to-U) as well as U-to-C and G-to-A, called "alternative mRNA editing", have been observed that could—at least in part—explain the remaining RNA-only variants[52]. A small fraction of RNA-derived neoantigen candidate variants in this cohort has not been covered by WES and could potentially be somatic mutations. Merlotti et al. showed recently that non-canonical splicing junctions between exons and transposable elements can lead to the formation of immunogenic neoantigens in non-small cell lung cancer patients[11], further supporting the relevance of RNA for targeted immunotherapy in cancer. Therefore, elucidating the nature of RNA variants and their role in cancer biology and immunotherapy is an important research area (reviewed in refs. 53–55) that might lead to new types of cancer treatment.

Neoantigen-based vaccines showed a limited clinical response in previous trials[5,56]. This might have been due to poor candidate selection or because of a dysregulated T-cell state in the treated patients. However, some efficacy has recently been observed using mRNA vaccination in melanoma and pancreatic cancer including also a combination with immune checkpoint inhibitors[4,57,58], suggesting that it is crucial to overcome the dysregulated T cell state for neoantigen vaccines to be efficacious. It will be important to understand subtle differences in vaccines and clinical protocols in order to understand the outcomes of these early trials. In addition, developing alternative strategies that engage non-dysfunctional T cells like neoantigen-specific TCR-T cell therapy is of great importance to treat patients that do not respond to immune checkpoint inhibition.

Taken together, our data identified a number of attractive cancer-associated and -specific canonical and non-canonical peptide antigens that have been partially shared by a significant portion of patients in our cohort. Moreover, we demonstrate the importance of RNA as a source for MS-based neoantigen identification in a large number of patients of this cross-disease cohort correlating with T-cell infiltration. Functionally active neoantigen-specific T cells could be identified only in a sub-cohort of these patients likely due to a severe dysfunctional state of these T cells. Therefore, immunotherapies focusing on the rescue of such T cells or targeting neoantigens with a non-dysfunctional repertoire including TCR-transgenic T cells may represent a valid immunotherapeutic option for a large number of cancer patients.

## Methods

### Human study

The study was approved by the institutional review boards (Ethics Commission of the Medical Faculty of Technical University Munich (protocol 193/17S) and Ethics Committee of the Medical Faculty of Heidelberg University (protocol S-206/2011)) and all patients provided written informed consent under these protocols. The study was conducted in accordance with the Declaration of Helsinki. Blood collection of healthy donors and the use of this material for the functional experiments in this study was approved by the Ethics Commission of the Medical Faculty of Technical University Munich (protocol 521/18 S-AS) and all participants provided written informed consent under this protocol.

### Primary human material and cell lines

An overview of all patients is given in Supplementary Table 1. The sex of patients was not considered in the study design and findings do not apply to one specific sex. Overall 14 female and 18 male patients were included. The patients' sex did not correlate with any experimental results in this study. Tumor tissue samples were collected from patients, who underwent tumor resection at the different DKTK partner sites. Immediately after resection, fresh tumor tissue was macroscopically dissected by an experienced pathologist and stored in PBS at 4 °C for transport or until processing. Additional tumor tissue was formalin-fixed and paraffin-embedded (FFPE). Before molecular analysis, tumor diagnosis was confirmed by a pathologist and tumor content was determined by an HE stain taken from the sample going to be used.

From the fresh tumor tissue, a part was snap-frozen and stored in liquid nitrogen (−196 °C) for later sequencing and mass spectrometry analysis.

From all remaining fresh tissue a single cell suspension was generated by mincing and digesting 0.2 g tissue pieces per tube for 90 min at 37 °C in 1 ml RPMI supplemented with 40 μL Enzyme H (Tumor dissociation kit human, Ref. 130-095-929, Miltenyi; Stock conc.), 5 μL Enzyme A (Tumor dissociation kit human, Ref. 130-095-929, Miltenyi; Stock conc.), 25 μL Hyaluronidase (Ref. H1115000, Sigma-Aldrich, 10 mg/mL stock), and 25 μL DNAse I (Ref. 11284932001, Sigma-Aldrich,

10 mg/mL stock). After digestion, the suspension and tissue pieces were meshed, and single cells were used for flow-cytometry analysis and FACS analysis.

Primary patient cells used in this study: For TIL generation, part of the fresh tumor tissues was minced and TILs were expanded for 2–3 weeks by cultivation with irradiated feeder PBMC, 1000 U/ml IL-2 (Ref. 200-02, PeproTech) and 30 ng/mL OKT-3 (kindly provided by Elisabeth Kremmer). A change of medium supplemented with 300 U/mL IL-2 was performed twice a week. After expansion for 2 weeks, TILs were frozen for later use in stimulation assays. PBMC from patients were isolated from whole blood by density-gradient centrifugation (Ficoll/Hypaque, Ref. L6115, Biochrom) immediately on receipt and frozen for later use in stimulation assays. Patients' T cells, derived from PBMCs or TILs, were cultivated in T-cell medium (TCM): RPMI 1640 (Ref. 11875093, Thermo Fisher Scientific) supplemented with Penicillin/Streptomycin (Pen/Strep) (Ref. 15140122, Thermo Fisher Scientific), 5% FCS (Ref. 26140079, Thermo Fisher Scientific), 5% human serum (HS), 10 mM Hepes (Ref. 15630080, Thermo Fisher Scientific), 10 mM MEM non-essential amino acids (Ref. 11140050, Thermo Fisher Scientific), 1 mM MEM sodium-pyruvate (Ref. 11360070, Thermo Fisher Scientific), 2 mM L-glutamine (Ref. 25030081, Thermo Fisher Scientific), and 16.6 μg/mL Gentamycin (Ref. A2712, Biochrom).

Cell lines used in this study: T2 (Ref. CRL-1992, American Type Culture Collection (ATCC)) and C1R cell lines (Ref. CRL-1993, ATCC) and lymphoblastoid cell lines (LCL) generated from patient samples (LCL IN-01, IN-03, IN-04, IN-08, IN-09, IN-11, IN-13, IN-18, IN-19, IN-22, IN-24, IN-33, and IN-37) and healthy donors (HD) (LCL HD04, HD06, HD07, HD08, and FM), purchased from ATCC (Daudi, Ref. CCL-213) or obtained from Steve Marsh (LCL CLA, IBW9, RSH, and SWEIG007) were used. Morphology and constant growth behavior of all cell lines were controlled periodically, and the absence of mycoplasma infection was routinely confirmed by PCR (Venor GeM mycoplasma detection kit, Ref. 11-1025, Minerva Biolabs). T2 and C1R were retrovirally transduced with the HLA restriction elements HLA-A6601 (C1R-A6601), B0702 (C1R-B0702), A0301 (T2-A0301), B5101 (T2-B1501), and B4402 (T2-B4402) as described before[6]. All target cell lines were maintained in complete RPMI (cRPMI): RPMI 1640 (Ref. 11875093, Thermo Fisher Scientific) supplemented with Pen/Strep (Ref. 15140122, Thermo Fisher Scientific), 10 mM MEM non-essential amino acids (Ref. 11140050, Thermo Fisher Scientific), 1 mM MEM sodium-pyruvate (Ref. 11360070, Thermo Fisher Scientific), 2 mM L-glutamine (Ref. 25030081, Thermo Fisher Scientific), and 10% FCS (Ref. 26140079, Thermo Fisher Scientific).

### Whole exome and RNA sequencing of patient material and analysis

**Extraction of nucleic acids.** DNA and RNA from tumor specimens and DNA from matched blood samples were isolated using the AllPrep DNA/RNA/miRNA Universal Kit (Ref. 80224, Qiagen). For formalin-fixed and paraffin-embedded (FFPE) samples, the AllPrep DNA/RNA FFPE Kit (Ref. 80234, Qiagen) was used. DNA from blood samples was isolated using the QIAsymphony DSP DNA Mini Kit (Ref. 937236, Qiagen) or the QIAamp DNA Blood Mini Kit (Ref. 51104, Qiagen). Quality control and quantification were done using a FilterMax F3 Multi-Mode Microplate Reader (Ref. F3, Molecular Devices) and a 4200 or 2200 TapeStation system (Ref. G2991BA, Agilent).

**Library preparation and target capture for whole-exome sequencing.** For whole-exome sequencing (WES) library preparation, 1.5 μg genomic DNA were fragmented to 150–200 base pair (bp; paired-end) insert size with a Covaris S2 device (Ref. 500217, Covaris), and 250 ng of Illumina adapter-containing libraries were hybridized with exome baits at 65 °C for 16 h. Exome capturing was performed using SureSelect Human All Exon in-solution capture reagents (Agilent). In case RNA was pooled for sequencing, V5 without UTRs was used (Ref.

5190–6209, Agilent) to reach a minimum average coverage of 80x for the tumor and 50x for the control. V5 with UTRs was used when DNA was sequenced alone (Ref. 5190–6213, Agilent).

**Library preparation for whole-genome sequencing.** Whole-genome sequencing (WGS) libraries were prepared using the TrueSeq Nano Library Preparation Kit (Ref. 20015965, Illumina) following the manufacturer's instructions.

**Library preparation for RNA sequencing.** RNA sequencing (RNA-seq) libraries were prepared using the TruSeq RNA Sample Preparation Kit v2 (Ref. RS-122-2001, Illumina) using the stranded protocol. Briefly, mRNA was purified from 1 μg total RNA using oligo(dT) beads, poly(A) + RNA was fragmented to 150 bp and converted into cDNA, and cDNA fragments were end-repaired, adenylated on the 3′ end, adapter-ligated, and amplified with 12 cycles of PCR. 2 The final libraries were validated using a Qubit 2.0 Fluorometer (Ref. Q33238, Life Technologies) and a Bioanalyzer 2100 system (Ref. G2939BA, Agilent).

**Whole-exome, whole-genome, and RNA sequencing.** Paired-end sequencing (2 × 150 bp) was performed with HiSeq X-Ten instruments (Illumina). Two lanes, each of tumor and control, were sequenced, yielding an average coverage of at least 70x for WGS cases. Paired-end sequencing (2 × 100 bp) was carried out with HiSeq 4000 (Illumina), pooling two patients' samples on one lane. From January 2017, RNA was sequenced separately with dual indexing in pools of three samples per HiSeq 4000 lane or multiplexed over several lanes to prevent adapter hopping. From October 2019, RNA was sequenced in pools of three to five samples per NovaSeq 6000 lane. The comparability of data has been validated.

**Mutation calling from exome and RNA sequencing data.** Mutation calling was performed on WES/WGS and RNA-Seq data for the identification of single-nucleotide variants and insertion/deletions for the indicated patients (Supplementary Table 1). Analysis of WES data was performed following the GATK Best Practice suggestions and utilizing the established analysis pipeline MoCaSeq[31], adapted for the human genome. After read trimming using Trimmomatic 0.38 (LEADING:25 TRAILING:25 MINLEN:50), bwa mem 0.7.17 was used to map reads to the human reference genome (GRCh38.p12, Ensembl release 91 [https://www.gencodegenes.org/]). Picard 2.18.26 and GATK 4.1.0.0 were used for post-processing (CleanSam, MarkDuplicates, BaseRecalibrator) using default settings. Somatic mutations were called using MuTect2 4.1.0.0[59]. SNVs and Indels ≤10 base pairs were annotated using SnpEff 4.3t, based on Ensembl 92.

For mutation calling from RNA-Seq, raw reads were trimmed using Trimmomatic (LEADING:25 TRAILING:25 SLIDINGWINDOW:10:25 MINLEN:50) and aligned to the human reference genome with STAR (2.6.0c). Mutations were called using Strelka2 (2.9.10) using the RNA option[60]. SNVs and Indels ≤10 base pairs were annotated using SnpEff 4.3t, based on Ensembl 92. De novo variant calling on tumor WES data was performed by comparison to PBMC WES data.

For variant calling on RNA-Seq data, positions sufficiently covered in WES with no evidence for the presence of germline SNVs/indels, were included as somatic. Furthermore, for positions where SNVs/indels were called only by Mutect2 or Strelka2, the threshold to include this SNV/indel in the second tissue sample was substantially lowered and did not require to be called separately by Mutect2/Strelka2.

Population SNPs with certain population allele frequency based on GnomAD[61] V2.1.1 (>1%, [https://gnomad.broadinstitute.org/]) and dbSNP (>5%, [https://www.ncbi.nlm.nih.gov/snp/])[62] were excluded (2020).

To calculate the tumor mutational burden (TMB), the first WES probe regions (±300 bp) with coverage above ten reads were identified. The TMB was then calculated as the number of genic/non-synonymous mutations overlapping with these regions divided by the total length of probe regions in megabases (Mb).

**Examination of variant prevalence in normal tissue RNA-seq data.** To confirm the tumor-specificity of potential neoantigens, the prevalence of the corresponding variants was examined in normal tissue RNA-seq samples. For this purpose, 10,269 samples across 30 different tissues were obtained from the Genotype-Tissue Expression (GTEx) project (January 2023, [https://gtexportal.org/home/])[35]. Variant detection was performed according to the tumor mutation calling procedure with Strelka2 (2.9.10) using the RNA option. The variant for each potential neoantigen candidate was then checked for its prevalence in the pool of variants from all GTEx samples. In case of a hit (at least one read), the number of altered/polymorphic reads in the normal tissue were annotated.

Additionally, for each tissue up to 100 samples were examined using Jvarkit (2021.10.13, c8e1dd6f7) with the findallcoverageatposition command, to extract the total number of reads at each mutation position. Prevalence for variants that were derived from positions that were covered in less than 5% of GTEx samples with at least three reads were considered not available.

### HLA typing

HLA typing was done from the available whole exome or whole-genome[63] sequencing data using the consensus of all xHLA (v1), BWAKit[64] (v1), and OptiType[65] (v1) using default settings. For confirmation, HLA typing was done on gDNA isolated from PBMC by targeted next-generation sequencing in selected patients (Zentrum für Humangenetik und Laboratoriumsdiagnostik, Martinsried, Germany).

### Immunoprecipitation of HLA complexes and liquid chromatography (LC)-MS/MS analysis of eluted peptides

Immunoprecipitation of HLA complexes, consequent elution and purification of peptide ligands was performed on indicated tumor samples (Supplementary Table 1) as previously described in ref. 6. Briefly, snap-frozen tumor tissue samples were placed in 5–7 ml of PBS with 0.25% sodium deoxycholate (Ref. S1827, Sigma-Aldrich), 1% octyl-β-D glucopyranoside (Ref. O8001, Sigma-Aldrich), 0.2 mM iodoacetamide (Ref. I5161, Sigma-Aldrich), 1 mM EDTA (Ref. E8008, Sigma-Aldrich), and 1:200 Protease Inhibitor Cocktail (Ref. P8340, Sigma-Aldrich) and mechanically dissociated with an ULTRA-TURRAX Disperser (Ref. 0003725000; IKA) for 10 s on ice, followed by 1 h incubation at 4 °C. The lysates were then cleared by centrifugation at 40,000 × g at 4 °C for 20 min and flowed through columns packed with protein-A Sepharose beads (Ref. 101041, Thermo Fisher Scientific) to deplete the endogenous antibodies. HLA class I complexes were immunoaffinity-purified from the cleared and antibody-depleted lysates on columns containing protein-A Sepharose beads covalently bound to 2 mg of the pan-HLA class I antibody W6/32 (purified from HB95 cells; ATCC) and eluted at room temperature with 0.1 N acetic acid. The eluted HLA-I complexes were then loaded onto Sep-Pak tC18 cartridges (Ref. WAT036820, Waters Corporation), and HLA-I peptides were separated from the complexes by elution with 30% acetonitrile (ACN, Ref. L010000, Sigma-Aldrich) in 0.1% trifluoroacetic acid (TFA, Ref. 74564, Sigma-Aldrich). Peptides were further purified using Silica C-18 column tips (Ref. 74-7226, Harvard Apparatus), eluted again with 30% ACN in 0.1% TFA and concentrated by vacuum centrifugation. Finally, HLA-I peptides were resuspended with 2% ACN in 0.1% TFA for LC-MS/MS analysis.

LC-MS/MS analysis was performed on an EASY-nLC 1200 system (Thermo Fisher Scientific) coupled online with a nanoelectrospray source (Thermo Fisher Scientific) to a QExactive HF-X mass spectrometer (Thermo Fisher Scientific). Peptides were loaded in buffer A (0.1% formic acid) on a 50 cm long, 75 μm inner diameter column, in-

house packed with ReproSil-Pur C18-AQ 1.9 μm resin (Ref. r119.aq., Dr. Maisch HPLC GmbH), and eluted during a 95 min linear gradient of 5–30% buffer B (80% ACN, 0.1% formic acid) at a flow rate of 300 nl/min. The mass spectrometer was operated in a data-dependent mode with the Xcalibur software (Thermo Scientific). Full MS scans were acquired at a resolution of 60,000 at 200 m/z and AGC target value of 3e6 with a maximum injection time of 80 ms. The ten most abundant ions with charge 1–4 were accumulated to an AGC target value of 1e5 for a maximum injection time of 120 ms and fragmented by higher-energy collisional dissociation (HCD). MS/MS scans were acquired with a resolution of 15,000 at 200 m/z and 20 s dynamic exclusion to reduce repeated peptide selection.

### Canonical peptidome analysis

For the identification of peptide sequences from the MS spectra, pFIND (v.3.1.5)[24] was used to match the reference protein database (Human Ensembl GRCh38, release 92, [https://www.gencodegenes.org/]) with general contaminants to the generated spectra files. The allowed precursor tolerance and fragment tolerance were set to 20 p.p.m. Parameters were defined to search for non-specifically digested peptides ranging from 8–15mers with a maximum mass of 1500 Da and N-terminal acetylation (42.010565 Da), methionine oxidation (15.994915 Da), and cysteine carbamidomethylation (57.021463 Da) were specified as variable post-translational modifications (PTMs). The FDR was set to 0.01 at the peptide spectrum match level.

### MHC-motif deconvolution

To assess the quality and purity of the MS-generated immunopeptidomic data, the identified peptide sequences were deconvoluted to the respective patient's HLA-allele by their binding motif using MHCMotifDecon-1.0[66,67]. Here, MHC binding predictions from NetMHCpan-4.1 (for MHC class I) are used to deconvolute and assign likely MHC restriction elements to MHC peptidome data. All identified peptide sequences with lengths of 8–15 amino acids and all HLA-A, B, and C alleles of each patient have been used for analysis applying standard setting as indicated on the website.

### Pipeline for the identification of patient-specific neoantigen candidates from MS data

In order to improve the identification of neoantigens we further developed our MS-based pipeline[6] for the analysis of this diverse patient cohort (Fig. 1). The following features have been integrated: (1) on the genetic level, mutation calling from RNA sequencing data has been accomplished using Strelka2[31]. Moreover, a refined algorithm for the translation of open reading frames (ORFs) in all three frames has been implemented to identify potential neoantigens from a large source of genetic aberrations (splice-site variants, intron-inclusions, non-coding variants, etc.). (2) On the proteomic level, pFind as a peptide calling tool[24] as well as the machine learning tool Prosit[21,22] have been included in the pipeline. (3) We additionally established a comprehensive post-processing filtering procedure, especially focusing on the exclusion of possible canonical peptides and SNPs. In detail, the subsequently described analysis steps have been performed.

### Generation of a custom database for MS-based identification of mutated peptides.

With the main goal to obtain mutated peptide sequences, mutations called from WES/WGS and RNA-seq were introduced into the wildtype transcript DNA sequences downloaded from biomart (v92, [https://www.ensembl.org/info/data/biomart/index.html]) and translated into peptide sequences using our custom code VCFtranslate[32]. Genes were included in the analysis without exceptions regarding the transcript biotypes. For non-protein-coding transcripts, ORFs enclosing the mutation site were determined by identifying the paired start and stop codons in all three reading frames. The same procedure was performed for protein-coding

transcripts in case of start/stop-loss/gain and frameshift mutations. Furthermore, for start/stop mutations, the coding sequence (CDS) was extended into the corresponding UTR. For mutations affecting splice donor or acceptor sites, the affected intron was included in the CDS and again checked for valid ORFs. Only mutations resulting in amino acid changes and within valid ORFs were considered. For every affected transcript, up to three ORFs enclosing the mutation site were translated into the corresponding mutated peptide sequence. Peptide sequences were then used together with the immunopeptidomics data from mass spectrometry.

### Identification of mutated peptides sequences from MS data.

For the identification of mutated HLA class I peptides, the reference protein database (Human Ensembl GRCh38, release 92, [https://www.gencodegenes.org/]) was searched together with the patient-specific customized databases containing the mutated sequences from step 1 using pFind (v.3.1.5)[24]. The allowed precursor tolerance and fragment tolerance were set to 20 p.p.m. Parameters were defined to search for non-specifically digested peptides ranging from 8–15mers with a maximum mass of 1500 Da and N-terminal acetylation (42.010565 Da), methionine oxidation (15.994915 Da), and cysteine carbamidomethylation (57.021463 Da) were specified as variable post-translational modifications (PTMs). The FDR was set to 0.05 at the peptide spectrum match level. After protein annotation, the pFind generated unfiltered peptide lists were (1) filtered for the FDR of 0.05 and used directly for further post-processing (pFind peptides) and (2) used unfiltered for subsequent rescoring and analysis by the Prosit pipeline (Prosit peptides)[21,22]. The rescoring method is extensively described in ref. 21. In brief, the unfiltered search engine output including decoys of pFind was used as input for the spectral intensity-based rescoring. Unprocessed MS2 spectra corresponding to the identifications were annotated with all matching b- and y-ions. Spectral comparison between predicted fragment ion intensities and experimental intensities was performed using the best-matching prediction settings and calculating previously described similarity measures (e.g., normalized spectral contrast angle). FDR estimation was performed using SVM Percolator 3.00[68]. All PSMs surpassing an FDR threshold of 5% were further considered for analysis.

Peptides identified by both approaches were combined and used for post-processing.

### Verification of mutated peptide sequences using synthetic peptides and predicted peptide properties.

Eighty-eight mutated peptide sequences were obtained as synthetic peptides from DGPeptidesCo Ltd. (>90% purity). Synthetic peptides were reconstituted in DMSO at a concentration of 100 μM and diluted at least 1:1000 in water with 0.1% FA. For LC-MS/MS analysis, peptides were combined in equimolar amounts and ~200 fmol per peptide was subjected to LC-MS/MS analysis using 0.1% FA in ultrapure water as sample buffer. LC-MS/MS analysis was performed on a U3000 RSLC nano-LC (Thermo Fisher Scientific) coupled online with a nanoelectrospray source (Thermo Fisher Scientific) to an Orbitrap Eclipse mass spectrometer (Thermo Fisher Scientific). Peptides were loaded in buffer A (0.1% formic acid) on a 40 cm long, 75 μm inner diameter column, in-house packed column (Dr. Maisch HPLC GmbH), and eluted during a 60 min linear gradient of 4–32% buffer B (80% ACN and 0.1% formic acid) at a flow rate of 300 nl/min. The mass spectrometer was operated in a data-dependent mode with the Xcalibur software (Thermo Scientific). Full MS scans were acquired at a resolution of 120,000 at 200 m/z and AGC target value of 4e5 with a maximum injection time of 50 ms. The most abundant ions were accumulated to an AGC target value of 1e5 for a maximum injection time of 50 ms and fragmented by higher-energy collisional dissociation (HCD) using varying collision energies to find matching fragmentation settings to the originally acquired data. MS/MS scans were acquired with a resolution of 15,000 at 200 m/z and 5 s

dynamic exclusion to ensure repeated peptide selection. RAW files were processed using MaxQuant 1.6.5.0 and a fasta file containing the 90 concatenated peptide sequences. For comparison of the endogenous MS/MS spectrum and the synthetic MS/MS spectrum, the spectra for the endogenous PSMs and the synthetic peptide PSMs were extracted using the Thermo RawFileReader library together with custom R scripts using the functionality of the package rawDiag (0.0.41)[69]. A pair-wise spectral comparison was performed by merging the two spectra with a tolerance of 20 ppm and then calculating the normalized spectral contrast angle (SA)[33]. The best spectral angle for each mutated peptide is reported in Supplementary Data 4. For the comparison of Prosit-predicted fragment ion intensities and the endogenous MS/MS spectra, the sequences were predicted using the Prosit HLA model. A pair-wise spectral comparison was performed by merging the endogenous spectra and the predicted fragment ion intensities with a tolerance of 20 ppm and then calculating the SA[33] using custom R scripts. The results of the verification are displayed in Fig. 7, a mirror plot visualization of the matching endogenous MS2 spectra, synthetic MS2 spectra, and predicted fragment ions per peptide sequence can be found in (Supplementary Data 5).

As a further verification criterion, the difference in retention time of the mutated peptide to the Prosit-predicted retention time was compared to all peptides identified. For this, the predicted indexed retention times (iRTs) of Prosit were aligned to the retention times in the raw file using LOESS-regression on a per-file basis. The difference of the predicted and experimentally determined retention times of all peptides and mutated peptides was used to classify identifications that either followed or did not follow the expected distribution (given all datapoints of the absolute RT errors ±8.56 min; Supplementary Fig. 11). Experimental RTs between 9 and 17 min that did not follow the expected distribution were considered deviations due to inaccuracies of the predictions at this range that was observed for the canonical peptides (Supplementary Fig. 11, yellow datapoints).

**Post-processing and filtering of neoantigen candidates and MHC binding prediction.** Peptides were filtered to remove contaminants and reverse sequences, which were only used to determine statistical cutoffs. In addition, the results were filtered for sequences identified exclusively in the custom mutated databases, and not in the Ensembl database, thus ensuring the peptide originating from a non-wild-type ORF.

Peptides harboring mutations (SNVs, In/Dels, multiple substitutions) within their sequence were directly taken as valid, whereas peptides not containing the mutations in the peptide sequence were further assessed. SNVs outside of the peptide sequence were excluded, whereas frameshift mutations upstream of a peptide or splice-site mutations were checked manually in BLAT[70] and were considered "mutated" or "non-canonical" if a peptide within a non-canonical frame or a retained intron was detected. The filtered potential neoantigens were then checked via an automated protein BLAST[71] search and peptides with more than two hits in the protein database were excluded while peptides with one to two hits were double-checked manually by literature research and excluded if necessary (April 2023). Additionally, three different peptide databases PeptideAtlas[72] [http://www.peptideatlas.org/], PepBank[73] [http://pepbank.mgh.harvard.edu/], and IEDB[74] [http://www.iedb.org/] were used to filter for already known (immunogenic) peptides.

After complete filtering the binding affinity of each neoantigen candidate was predicted by using two different algorithms, NetMHC (v.4.0)[36] and MHCflurry (v.1.6.0) (models class1)[37] run with Python (v.3.6), and the best binding allele according to predicted affinity or percentile rank was determined for each algorithm.

**Flow-cytometry analysis of tumor single cells and FACS sort**
For flow-cytometry analysis, up to 0.5 Mio alive single cells from the digested tumor tissue have been used per panel and isotype controls.

Cells were first incubated in 50 μL human serum (HS) for 20 min for blocking unspecific binding. Subsequently, ethidium monoazide bromide (EMA, 1:500, Ref. E1374, Thermo Fisher Scientific) was added for live-dead staining to the HS and incubated for 10 min on ice in the dark and 10 min on ice in the light. After washing, the respective antibodies or the isotype control antibodies were added (1:50 dilution) and stained for 20 min on ice in the dark. The following antibodies were used: CD45-PerCP-Cy5.5 (clone HI30, Ref. 564105, BD, RRID:AB_2744405), CD3-AF700 (clone UCHT1, Ref. 300423, BioLegend, RRID:AB_493740), CD8-APCH7 (clone SK1, Ref. 560179, BD, RRID:AB_1645481), CD4-V450 (clone SK3, Ref. 651849, BD, RRID:AB_2870340), CD45RA-BV510 (clone HI100, Ref. 304141, BioLegend, RRID:AB_2561384), CD62L-PE (clone DREG-56, Ref. 560966, BD, RRID:AB_2033966), CD366-BB515 (anti-TIM-3, clone 7D3, ref. 565568, BD, RRID:AB_2744368), CD279-PE-Cy7 (anti-PD-1, clone EH12.2H, Ref. 329917, BioLegend, RRID:AB_2159325), CD223-APC (anti-LAG-3, clone 3DS223H, Ref. 17-2239-42, eBioscience, RRID:AB_2573186), CD103-FITC (clone Ber-ACT8, Ref. 550259, BD, RRID:AB_393563), HLA-DR-APC (clone G46-6, Ref. 559866, BD, RRID:AB_398674), and CD45-APC-H7 (clone 2D1, Ref. 560274, BD, RRID:AB_1645480). All antibodies were validated in the RRID registry (https://scicrunch.org/resources/Antibodies/search?l=&q=%2A, RRID identifiers are listed above). Appropriate isotype controls for each antibody were used as a negative control. After staining, cells were washed and fixed with paraformaldehyde (PFA, 1%, Ref. 158127, Sigma-Aldrich) and stored at 4 °C for later analysis. Measurements were performed on an LSR II (BD) and anti-IgG beads (Ref. 130-047-501, Miltenyi), as well as unstained cells, were used for single stains and instrument set-up. Voltages were adapted to the autofluorescence of each patient tumor and all possible events were measured using FACS DIVA software. All steps were carried out on the ice and as quickly as possible to minimize changes in cell viability and marker expression. Data analysis and compensation was performed using FlowJo V10.7.1 and the gating strategy was kept consistent for every sample depending on the panels analysed (gating strategy see Supplementary Fig. 2a).

For sorting of CD8+ T lymphocytes (sorting strategy see Supplementary Fig. 3), min. 5–10 Mio cells were taken from the digested tumor sample/single cell suspension (when enough cells were available). Cells were blocked with 200–500 μL HS depending on the cell numbers for 20 min on ice in the dark. After washing 2 μL/1 Mio cells of the respective antibodies, CD8-PE-Cy7 (clone RPAT-8, Ref. 557746, BD, RRID:AB_396852) and CD45-APC (clone J33, Ref. IM2473, Beckman Coulter, RRID:AB_130783), and 7-amino-actinomycin D (7-AAD, Ref. A1310, Thermo Fisher Scientific) for live-dead staining were incubated in 100–200 μL FACS Buffer for 30 min on ice in the dark. All antibodies were validated in the RRID registry (https://scicrunch.org/resources/Antibodies/search?l=&q=%2A, RRID identifiers are listed above). After washing cells were resuspended in 1 mL/10 Mio cells FACS buffer, filtered and directly used for sorting on a FACSAria III (BD). Single stains were generated using anti-IgG microbeads (Ref. 130-047-501, Miltenyi) according to the manufacturer's instructions and were used together with unstained cells and 7-AAD-only stained cells for on-device compensation. Alive-SingleCells-CD45+-CD8+ cells were sorted into pre-cooled tubes containing RPMI. Sorted cells have been pelleted and resuspended in 300 μL RNA Protect (Ref. 76104, Qiagen), snap-frozen, and stored in liquid nitrogen (−196 °C) for later mRNA sequencing analysis. All steps have been carried out on the ice and as quickly as possible to minimize changes in cell viability and marker expression.

**Bulk RNA sequencing of sorted cells and analysis**
Paired-end sequencing (2 × 75 bp) was performed on a NextSeq 500 (Illumina) with SMART-Seq Stranded Kits (Ref. 634862, Takara Bio) to reach at least 50 Mio. raw reads per sample. The raw sequencing data were processed with Trimmomatic version 0.36[75]. Trimmed reads were acquired by removing Illumina TruSeq3 adapters and bases at the start

and end of each read, for which the phread score was below 25. Further reads were clipped if the average quality within a sliding window of 10 fell below a phread score of 25. Conclusively reads smaller than 50 bases were removed. For mapping and counting, the human gene annotation release 29 and the corresponding genome (GRCh38.p12, [https://www.gencodegenes.org/]) were derived from the GENCODE homepage[76]. STAR version 2.7.5b[77] was used to map the trimmed sequencing data to the reference genome, with the parameters adapted from protocol recommendations[78]. Mapped reads were deduplicated with bamUtils v1.0.14[79] and featureCounts v.1.6.3[80] was used to assign and summarize reads to genes while ignoring multi-mapping, multi-overlapping, and duplicated reads. The resulting raw count matrix was imported into R v4.0.5 and lowly expressed genes were subsequently filtered out. Prior to differential expression analysis with DESeq2 v1.18.1[81], dispersion of the data was estimated with a parametric fit using the Survival as explanatory variable. Shrunken log2 fold changes were calculated afterward with the apeglm method[82] and used as ranking criteria for the pathway analysis with GSEA in the preRanked mode[83]. The Hallmark and Gene Ontology gene set definitions from MsigDB v7.4[84,85] [https://www.gsea-msigdb.org/gsea/msigdb] were used for GSEA. The presence of neoantigen candidate variants in CD8+ TIL RNA-seq data was investigated as was described for the analysis of GTEx samples.

## Generation of lymphoblastoid cell lines as autologous target cells

For the generation of patient-derived LCL, the first potent Epstein-Barr virus (EBV) supernatant was generated from B95-8 cells (provided by Ulrike Protzer). Therefore, 1 Mio cells per mL were stimulated in cRPMI (see Methods—primary human material and cell lines) with 20 ng/mL PMA (Ref. P1585, Sigma-Aldrich) for 1 h at 37 °C, subsequently washed three times and cultured at a concentration of 1 Mio cells per mL in fresh cRPMI. After 3 days the supernatant was harvested, filtered with a 0.45 μm sterile filter, and stored at −80 °C for up to 1 year. Afterward, this supernatant was used for the infection and immortalization of patient-derived B cells from PBMC samples. Therefore, up to five Mio PBMCs were incubated in 1 mL RPMI with 1 mL EBV supernatant for 2 h at 37 °C, following the addition of further 1 mL cRPMI supplemented with Cyclosporine A (Ref. 32425, Sigma-Aldrich) to a final concentration of 1 μg/mL and culture in cell culture flasks at 37 °C. Cells were split once clusters were visible and/or medium color changed and expanded at 0.3–0.6 Mio cells per mL until enough cells were available for freezing or direct use in experiments.

## Immunogenicity assessment of identified peptide ligands

Recall antigen-experienced T cell responses to selected peptides were investigated as previously described with modifications[6,86]. In brief, up to 1 Mio PBMCs or TILs per well from each patient were used for in vitro screening. For peptide stimulation on day 0, 1 μM of each synthetic peptide (>90% purity, DGPeptidesCo Ltd.) was added to the culture along with 0.5 ng/ml Interleukin (IL)-7 (Ref. 200-07, Peprotech), 50 ng/ml Tumor necrosis factor (TNF)-α (Ref. 300-01 A, Peprotech), and 10 ng/ml IL-1β (Ref. 200-01B, Peprotech). As positive control, T cells have been non-specifically stimulated with 0.5 ng/μL phorbol-12-myristate-13-acetate (PMA, Ref. P1585, Sigma-Aldrich) and 1 ng/μL Ionomycin (Ref. I9657, Sigma-Aldrich). After 24 h of peptide stimulation, 100 μL supernatant was collected for later ELISA analysis, and cells were either used for direct overnight ELISpot analysis as previously published or enriched for specifically activated T cells using a CD137+-based magnetic isolation[87]. CD137-expressing activated cells were isolated and enriched using the human CD137 MicroBead Kit (Ref. 130-093-476, Miltenyi) according to the manufacturer's instructions. Enriched cells were taken into a culture in T cell medium (TCM, see

Methods—primary human material and cell lines) supplemented with 5 ng/mL IL-7 (Ref. 200-07, Peprotech), 5 ng/mL IL-15 (Ref. 200-15, Peprotech), 30 U/mL IL-2 (Ref. 200-02, Peprotech), and 30 ng/mL OKT-3 (kindly provided by Elisabeth Kremmer) along with 1 Mio irradiated (30 Gray) feeder PBMC. Enriched cells were cultured for 12 days and fed by adding IL-7 and IL-15 twice per week and IL-2 once per week. Non-enriched cells were cultured and expanded in TCM supplemented with 5 ng/mL IL-7 and 5 ng/mL IL-15 and fed twice per week. For assays using healthy donor PBMCs the protocol without enrichment was followed and a different HLA-matched donor for each peptide was selected based on the affinity predictions performed by NetMHC 4.0[36] and MHCflurry[37] run with Python (v3.6), where possible.

After 13 days of expansion, reactivities of T cells to the synthetic peptide ligands was assessed by specific interferon (IFN)-γ release by ELISpot assay. As antigen-presenting target cells for the second stimulation on day 13, either an autologous lymphoblastoid cell line (LCL) derived from the same patient or HLA-matched LCL, HLA-transduced T2, or C1R cells were used. The target cells were pulsed for 2 h with either the selected mutated peptide or a control peptide prior to co-culture with the T cells (in duplicate or triplicate according to available cell numbers). The co-cultures were performed with an effecter-to-target ratio of 2:1 using 20,000 pre-stimulated T cells and 10,000 pulsed target cells per well. ELISpot plates (Ref. MAHAS4510, Merck Millipore) were coated with an IFN-γ capture antibody (1-D1K, 1:100 dilution, Ref. MAB-3420-3-250, Mabtech, RRID:AB_907283) at 4 °C overnight prior to the co-culture, development was performed with an IFN-γ-detection antibody (7-B6-1-biotin, 1:500 dilution, Ref. MAB-3420-6-1000, Mabtech, RRID:AB_907272) and Streptavidin-HRP (Ref. MAB-3310-9, Mabtech). ELISpot plates were read out on an ImmunoSpot S6 Ultra-V Analyzer using Immunospot software 5.4.0.1 (CTL-Europe).

We defined the reactivity by the spot counts on day 13 comparing the mean spots from the mutated peptide condition with the mean spots from the control peptide condition and set the threshold to a ratio above 2, meaning the mutated peptides elicited an IFN-γ response twice as many T cells compared to the control, and a difference of spots above 50, which is defined as the background threshold.

## Statistical analysis

The correlations of two distinct parameters were assessed using Spearman's rank correlation coefficient. For the correlation of the numbers of DNA variants and RNA variants, all samples with both analyses were included while for the correlation of phenotypic data with peptidomics data, only one representative tumor sample from each patient was included for analysis (ImmuNEO core cohort see Supplementary Table 1) has been used to circumvent bias due to multiple metastasis available for some patients. Two-tailed Mann–Whitney U-test with Benjamini–Hochberg correction was used to compare the immunophenotyping data of the TME and the size of the immunopeptidome for validated immunogenic and non-immunogenic neoantigen candidates.

Kaplan–Meier curves with log-rank test and Cox's proportional hazards model was used to evaluate the overall survival (OS) since tumor resection between a high and low patient group of ImmuNEO patients. For continuous parameters, groups were divided by the median into high (above median) and low (below median) groups. For relative parameters (0–100%), patients were divided into a high group with fractions above 50% and a low group with fractions below 50%. Here, only one representative tumor sample from each patient (ImmuNEO core cohort see Supplementary Table 1) has been used to circumvent bias due to multiple metastasis available for some patients.

## Reporting summary

Further information on research design is available in the Nature Portfolio Reporting Summary linked to this article.

## Data availability

All multiomic data have been deposited to public repositories. The mass spectrometry proteomics data have been deposited to the ProteomeXchange Consortium via the PRIDE[88] partner repository with the data set identifier PXD037655. The WES/WGS and RNA-seq data for variant calling, and the RNA-seq data from sorted CD8[+] TILs have been deposited to the European Genome-phenome Archive (EGA)[89] with the study identifier EGAS00001006706 and are available on request from the associated Data Access Committee (hipo_daco@dkfz-heidelberg.de) due to them containing patient information under controlled access. Access will be granted to commercial and non-commercial parties according to patient consent forms and data transfer agreements. We have an institutional process in place to deal with requests for data transfer and aim for a rapid response time. The publicly available RNA-seq data from 10,269 samples across 30 different normal tissues were obtained from the Genotype-Tissue Expression (GTEx) project (January 2023, [https://gtexportal.org/home/])[35]. The remaining data were available within the Article, Supplementary Information, Supplementary Data, or Source Data file. Source data are provided with this paper.

## Code availability

The source code of VCFtranslate[32] is freely available at GitHub: https://doi.org/10.5281/zenodo.7965941.

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

## Acknowledgements

We thank G. Swinerd and S. Mall for the preliminary organization of sample collection, the measurement of some patient samples, and the preliminary assembly of flow-cytometry antibody panels. We thank the virology department of MRI of U. Protzer for technical support and advice with ELISpot measurements and analysis. Many thanks to J. Gagneur for the helpful discussions. We especially thank A. Stelzl for logistics organization, measurement of patient samples, and experimental support. We thank the NCT Molecular Precision Oncology Program for technical support and funding through project number 21. A.M.K. was supported by the Deutsche Forschungsgemeinschaft (DFG, German Research Foundation)—SFB-TRR 338/1 2021–452881907 (to A.M.K./A03) and the German Cancer Consortium (DKTK) Joint Funding Program (ImmuNEO MASTER). N.v.B. was supported by the DFG (Project ID 386260575) and the German Federal Ministry of Education and Research (BMBF) within the OUTLIVE consortium (Project ID 01KD2103A). S.F. was supported by the NCT Molecular Precision Oncology Program and the DKTK Joint Funding Program.

## Author contributions

C.T., S.F., W.W., M.H., and A.M.K. conceived the study. C.T., N.d.A.K., M.P., P.S., J.U., F.S.D., E.B., K.P., E.S., B.Kl., M.M., and A.M.K designed and/or performed experiments. C.T., N.d.A.K., M.P., S.L., P.S., C.v.F., J.U., G.Z., M.W., D.P.Z., T.E., K.G., B.Ku., M.M., R.R., M.H., and A.M.K. analyzed data. C.T., N.d.A.K., M.H., G.Z., S.L., P.S., T.E., M.W., D.P.Z., S.U., K.G., B.H., B.Ku., and R.R. performed statistics and bioinformatics. M.Boxberg., K.St., J.S.-H., S.O., N.v.B., S.B., M.Boerries, P.J.J., K.Sc., I.D., F.B., H.F., D.R., K.P., and W.W. provided patient material. C.T., M.H., and A.M.K. wrote the manuscript.

## Funding

## Competing interests

The authors declare no competing interests.

## Additional information

Celina Tretter [1,2,37], Niklas de Andrade Krätzig [3,4,5,37], Matteo Pecoraro [6,7], Sebastian Lange [3,4,5], Philipp Seifert [2], Clara von Frankenberg [2], Johannes Untch [2], Gabriela Zuleger [2], Mathias Wilhelm [8,9], Daniel P. Zolg [8], Florian S. Dreyer [2], Eva Bräunlein [2], Thomas Engleitner [4,5], Sebastian Uhrig [10,11], Melanie Boxberg [1,12], Katja Steiger [1,12], Julia Slotta-Huspenina [12], Sebastian Ochsenreither [13,14,15], Nikolas von Bubnoff [16,17,18], Sebastian Bauer [19,20], Melanie Boerries [16,17], Philipp J. Jost [1,2,21,22], Kristina Schenck [1,2], Iska Dresing [1,2], Florian Bassermann [1,2,4], Helmut Friess [23], Daniel Reim [23], Konrad Grützmann [24,25,26], Katrin Pfütze [10], Barbara Klink [24,27], Evelin Schröck [24,27,28,29,30,31,32], Bernhard Haller [33], Bernhard Kuster [1,8,34], Matthias Mann [6], Wilko Weichert [1,12], Stefan Fröhling [10,35], Roland Rad [1,3,4,5], Michael Hiltensperger [1,2,38] & Angela M. Krackhardt [1,2,4,36,38] ✉

[1]German Cancer Consortium (DKTK), partner site Munich and German Cancer Research Center (DKFZ), Heidelberg, Germany. [2]Technical University of Munich, TUM School of Medicine, Klinikum rechts der Isar, IIIrd Medical Department, Munich, Germany. [3]Technical University of Munich, TUM School of Medicine, Klinikum rechts der Isar, IInd Medical Department, Munich, Germany. [4]Technical University of Munich, TUM School of Medicine, Center for Translational Cancer Research (TranslaTUM), Munich, Germany. [5]Technical University of Munich, TUM School of Medicine, Institute of Molecular Oncology and Functional Genomics, Munich, Germany. [6]Department of Proteomics and Signal Transduction, Max Plank Institute of Biochemistry, Munich, Germany. [7]Institute for Research in Biomedicine, Università della Svizzera italiana, Bellinzona, Switzerland. [8]Technical University of Munich, TUM School of Life Sciences, Chair of Proteomics and Bioanalytics, Freising, Germany. [9]Technical University of Munich, TUM School of Life Sciences, Computational Mass Spectrometry, Freising, Germany. [10]German Cancer Consortium (DKTK), partner site Heidelberg and German Cancer Research Center (DKFZ), Heidelberg, Germany. [11]Molecular Precision Oncology Program, NCT Heidelberg, Heidelberg, Germany. [12]Technical University of Munich, TUM School of Medicine, Klinikum rechts

der Isar, Institute of Pathology, Munich, Germany. [13]German Cancer Consortium (DKTK), partner site Berlin and German Cancer Research Center (DKFZ), Heidelberg, Germany. [14]Charité Comprehensive Cancer Center, Charité – Universitätsmedizin Berlin, Berlin, Germany. [15]Department of Hematology, Oncology and Tumor Immunology, Charité – Universitätsmedizin Berlin, Berlin, Germany. [16]German Cancer Consortium (DKTK), partner site Freiburg and German Cancer Research Center (DKFZ), Heidelberg, Germany. [17]Institute of Medical Bioinformatics and Systems Medicine (IBSM), Medical Center - University of Freiburg, Faculty of Medicine, University of Freiburg, Freiburg, Germany. [18]Department of Hematology and Oncology, Medical Center, University of Schleswig Holstein, Campus Lübeck, Lübeck, Germany. [19]German Cancer Consortium (DKTK), partner site Essen and German Cancer Research Center (DKFZ), Heidelberg, Germany. [20]Department of Medical Oncology and Sarcoma Center, West German Cancer Center, University Hospital Essen, Essen, Germany. [21]Clinical Division of Oncology, Department of Internal Medicine, Medical University of Graz, Graz, Austria. [22]University Comprehensive Cancer Center Graz, Medical University of Graz, Graz, Austria. [23]Technical University of Munich, TUM School of Medicine, Klinikum rechts der Isar, Department of Surgery, Munich, Germany. [24]German Cancer Consortium (DKTK), partner site Dresden and German Cancer Research Center (DKFZ), Heidelberg, Germany. [25]Core Unit Molecular Tumor Diagnostics (CMTD), NCT Dresden, Dresden, Germany. [26]Institute for Medical Informatics and Biometry, Faculty of Medicine, TU Dresden, Dresden, Germany. [27]Institute for Clinical Genetics, University Hospital Carl Gustav Carus at the Technische Universität Dresden, Dresden, Germany. [28]ERN GENTURIS, Hereditary Cancer Syndrome Center Dresden, Dresden, Germany. [29]National Center for Tumor Diseases Dresden (NCT/UCC), Dresden, Germany. [30]Faculty of Medicine and University Hospital Carl Gustav Carus, Technische Universität Dresden, Dresden, Germany. [31]Helmholtz-Zentrum Dresden-Rossendorf (HZDR), Dresden, Germany. [32]Max Planck Institute of Molecular Cell Biology and Genetics, Dresden, Germany. [33]Technical University of Munich, TUM School of Medicine, Klinikum rechts der Isar, Institute of AI and Informatics in Medicine, Munich, Germany. [34]Technical University of Munich, TUM School of Life Sciences, Bavarian Biomolecular Mass Spectrometry Center (BayBioMS), Freising, Germany. [35]Division of Translational Medical Oncology, National Center for Tumor Diseases (NCT) Heidelberg, German Cancer Research Center (DKFZ), Heidelberg, Germany. [36]Malteser Krankenhaus St. Franziskus-Hospital, Flensburg, Germany. [37]These authors contributed equally: Celina Tretter, Niklas de Andrade Krätzig. [38]These authors jointly supervised this work: Michael Hiltensperger, Angela M. Krackhardt. ✉e-mail: angela.krackhardt@tum.de

