## [Peer Review File · Nature Communications]

Proteogenomic analysis reveals RNA as a source for tumor-agnostic neoantigen identificationREVIEWER COMMENTS

Reviewer #1 (Remarks to the Author): with expertise in HLA-immunopeptidomics

Synopsis

The authors present an interesting study proposing HLA presented, RNA-derived variants as an immunologically relevant and potentially more prevalent source of neoepitopes than “classical” genomic somatic mutations. The authors leverage a proteogenomic workflow consisting of DNA- and RNA sequencing combined with HLA peptidomics across a set of 32 samples from 25 tumor types, resulting in the identification of 91 candidate neoantigens, the majority of which is derived from RNA variants. Immunogenicity testing of a subset of peptides and donors suggests memory T cell recognition of around 30% of these variant peptides.

Major Comments

While the study is appropriately sized and designed for an exploratory assessment of RNA-derived, HLA presented variants across tumor types, the central claims of tumor association/specificity of said variants requires further validation and their unambiguous molecular verification is mandatory. These two central prerequisites must be met prior to publication in Nature Communications.

The authors use relaxed criteria for variant calling and mass spectrometric identification which is an appropriate way to improve sensitivity, as long as sequence verification of the variant peptides is performed. Due to the degeneracy of TCR recognition, the observed immunogenicity of some of the variant peptides alone cannot be deemed sufficient verification. Sequence confirmation by co-elution using spiked-in stable isotope labeled peptides as described in (Chong et al. 2020, <https://doi.org/10.1038/s41467-020-14968-9> and Fritsche et al., 2021, <https://doi.org/10.1016/j.mcpro.2021.100110>) should be performed to ultimately provide reliable frequency estimates for RNA- vs DNA derived variant peptides. This should be feasible in light of the limited number of variant peptide candidates and would help avoid inaccurate prevalence estimates as previously observed in the field of HLA peptidomics in the context of proteasomally spliced peptides (Lichti et al., 2022, DOI: 10.1158/2326-6066.CIR-21-0727).

The claim of tumor association/specificity of the observed RNA variants cannot be supported by the provided data. Comprehensive assessment of variant expression levels across various, ideally matched, normal tissues are required to ascertain tumor-specificity and support the classification as RNA-derived neoantigens or to verify tumor-associated RNA-overediting and classify the peptides accordingly (see Zhang et al, 2018 for examples of RNA editing events in normal tissue, <https://doi.org/10.1038/s41467-018-06405-9>). This may be addressed by processing publicly available RNAseq repositories of normal tissues such as GTEx (www.gtexportal.org). Prematurely annotating variants as targets while not sufficiently screening normal tissues poses severe risks for on-target toxicities in later clinical use of such targets.

Minor Comments

- DNA vs RNA variant calling (Figure 1, also in lines 539-558): Different callers were used for WES/WGS and RNAseq. Given that Strelka2 supports both DNAseq and RNAseq variant calling, it may have been beneficial to only use a single caller/pipeline to avoid calling bias which may affect the downstream classification of variants as DNA/RNA-only.
 - o The filtering strategy may have missed some known/annotated SNPs. For instance the proposed neoantigen ALSGHLETL is listed in dbSNP (rs17845226) and thus less likely a somatic mutation. To prove that the additional RNA-based events are bona-fide somatic mutations and not false positive calls requires an assessment of how reliably WES-confirmed mutations can be called by RNAseq. Even for WES high numbers of false positives for de novo calls are expected
- Summary statistics of DNA/RNA variants vs canonical reads would be highly informative to compare coverage, minor allele fractions, number of reads, normalized expression
- The majority of peptides listed in Supplementary Table S3 are misleadingly called tumor-associated peptides, yet the only evidence is the assignment in Protein atlas as “cancer-gene” based on COSMIC mutations in the protein of origin and annotation of that protein as plasma biomarker in cancer. No evidence for over-expression in RNA or on peptide-level in cancer tissues is provided.

Furthermore, 14 of 18 of these peptides are found in the HLA Ligand Atlas (<https://hla-ligand-atlas.org>) thus confirming presentation of these peptides on healthy tissues.

- HLA peptidomics: FDRs for identification of variant peptides are very relaxed thus clearly calling for peptide verification prior to downstream immunogenicity testing. Variant peptides from two different search pipelines (pFind @5% FDR & pFind no FDR + Prosit/Percolator spectral rescoreing @5% FDR) were utilized to derive a combined list of putative variant peptides. Apart from global FDR potentially being inaccurate for low prior probability subsets of peptides, the combination of two sets of search results for the same raw data further inflates potential false positive IDs. An indication of potential inflation in the non-canonical IDs is given by the low number of HLA binders in table S4 (9 of 23) compared to S3.

- o The authors have previously presented predicted retention time as an important parameter apart from spectral correlation/angle for improving specificity and sensitivity of peptide rescoring pipelines (<https://www.msaid.de/inferys>). Retention time was not used as a scoring parameter in this study and should be included, at least for post hoc assessment of variant peptide IDs, e.g. as performed in Chong et al. 2020, <https://doi.org/10.1038/s41467-020-14968-9> by correlation of predicted vs experimental retention times

- o Summary statistics such as FDR, spectral correlation, %predicted binders, delta predicted RT should be provided to compare variant vs canonical peptide IDs

- o Relevant information for reproduction of peptide identifications are missing (e.g. mass tolerance used for MS1 and MS2, discrimination between static and dynamic PTMs, considered charge states, ...). MS2 spectra of the proposed variant peptides are not provided.

- Fig. S5A: "Peptides per gram tumor" should not be plotted as shown here as it suggests linear correlation of tumor input and peptides present / identified, which is not (necessarily) the case. Extrapolation is misleading.

Reviewer #2 (Remarks to the Author): with expertise in proteogenomics, RNA editing

The paper by Tretter et al. is focused on tumor neoantigen identification on the level of HLA-bound peptides in 32 patients partly already characterized in a specific MASTER program. The novelty was that authors considered specifically neoantigens originated from RNA variants. They were shown to be less variable than mutation-associated genomic mutations and, in further perspective, are considered as components for cancer vaccines.

The results compose an important multiomic dataset for further use. I would not say that they may have immediate clinical importance, as neoantigen vaccines still persist on the stage of I phase clinical trials and may never leave them for a wider practice (which authors admit in the discussion). At the same time, the new data presented have substantial basic significance. I am, thus, generally positive about this paper.

It is impossible to check all the huge data array presented here for their quality during the review. Let's just trust the author's reputation in this respect. Which could be done for improvement of the manuscript, it is to clarify some details of study in the text and to explain some decisions in data processing. Below I present some specific comments for authors' consideration. Most of them relate to the peptidomic part as nucleic acid pipelines and methods are better established and standardized.

1. DNA vs. RNA neoantigens and nucleic acid polymorphism. The important result which, in spite it was expected, was that DNA variants are more variable, i.e. personalized than RNA variants. Please discuss it more elsewhere why we observe this result. In the present form, the discussion is focused on future clinical applications that could not happen. Less attention is paid to the basic cancer biology.

2. In relation to the previous comment, please discuss more the biology of RNA variants, including those which compose neoantigens. The majority of them are A-to-G and are mentioned in Rediportal which means they are, most likely, products of ADAR RNA editing. How could occur the rest of them? This was not properly discussed.

3. Not all RNA neoantigens elicit immune response in ELISPOTA tests. We could say even that many of them failed to do so. Maybe, some of them are not NEO antigens. As A-to-I RNA editing may happen before thymus development (and a formation of the repertoire of immunological tolerance), some of these antigens formed via RNA editing could be presented to the immune system. Maybe, this could be mentioned in Results or Discussion sections.

4. Identification of variant-containing peptides (HLA ligands). This stage is technically problematic because the presence in the preparation of non-relevant peptides with low affinity to HLA adds difficulty as well as false assignments of PSMs which are not constrained by trypsin cleavage. Some questions arose to this parts that would be better replied in the revised paper.

4.1. Why so called wild-type, consensus genome-encoded peptides were analyzed (and results described) separately from DNA and RNA mutated peptides? Theoretically, if we take a customized (personalized) database for each sample which contains consensus and personalized sequences, we may get right PSMs for both "wild-type" and variant peptides.

4.2. I could not find in results a direct indication of how many "wild-type" peptides of identified did not dock eventually to HLA-I of corresponding alleles for each sample. This number is important for the perception of the results. (I could miss something in Supplementary Files, if so, I am sorry for this).

4.3. If I understood correctly, for variant HLA-I-bound peptide search, all variants which could potentially be coding (even outside recognized human ORFs) were read by 3 potential reading frames, admixed to consensus genomic database and then searched. An additional measure which made the output more filtered was the prediction with Prosit. The question is, how many records were added to the search database via this 3-frame reading. How much do they outnumber a conventional database? It was shown, for example, that addition of whole COSMIC (all cancer mutations known) to the consensus database led to the so called database inflation and dramatically increased a real FDR in spite of 1% FDR conservation by the software (in the target-decoy method). At least, deduced 3-frame variants must be filtered separately from genomic peptides which are much more likely to be identified (so called group-specific FDR), which was not done.

5. Data availability. A significant feature of the paper is the high-quality multiomic data generated. Now these data are available from the site of one of the institutes involved in the study. It would be preferable to make them available from one of public repositories. Unfortunately, data on institute's sites tend to stay neglected and disappear after finishing someone's postdoc and PhD contracts.

Minor comment:

In the text, a genetic variant as a term was applied to both DNA and RNA variants. I do not think this is exact, as RNA variant is not genetic, but phenotypic. I think it may be modified to avoid confusion. Further, 'wild-type' definition sounds weird in respect of the human data (yes I also used it, but it is really argotic expression, in that case, not scientifically exact).

Reviewer #3 (Remarks to the Author): with expertise in HLA-immunopeptidomics, Ag presentation

The authors found that considerable numbers of genetic variants were generated at the RNA level (e.g. by A-to-I RNA editing) across a variety of tumor types. Mutated peptides were translated and some were displayed by HLA class I molecules. Importantly, patient CD8+ T cells recognized those mutated peptides, suggesting that RNA variants, which are likely intact at the DNA level, served as a novel source of antigens potentially leading to anti-tumor immune responses. In addition, variant calling using RNA enabled to detect non-coding RNA-derived neoantigen candidates. These findings are interesting. To my knowledge, it is still difficult to identify real neoantigens that induced spontaneous patient T cell responses in vivo. This may be due to lack of methodology or the presence of yet unexplored sources of neoantigens. Therefore, the findings in this study are of importance and contribute to the research field of tumor antigens and T cell surveillance. Here, I have two questions.

1. Even if RNA variants gave rise to immunogenic antigens and could induce reactive T cell responses, it is still unclear if target antigens are tumor-specific. In contrast to somatic DNA mutation, RNA editing can occur and possibly lead to HLA presentation of same immunogenic antigens in normal cells. In this case, antigens may serve as autoantigens. The authors demonstrated antigen recognition by TILs in Fig 6A, which suggested the tumor specificity of this particular peptide; however, most identified antigens were recognized only by autologous PBMCs across patients. Do authors have additional evidence showing their tumor specificity?

2. A single tumor mass should be heterogeneous regarding an RNA WT/variant expression ratio. Is it possible to estimate clonality of each RNA variant within a tumor and clarify if the clonality is associated with any biological consequences (e.g. HLA presentation or induction of T cell responses)?

Rebuttal letter

We would like to thank all reviewers for their detailed assessment of our study and the constructive feedback. We believe that the revision process has significantly improved our manuscript and that the additional validation analyses that were suggested by the reviewers solidified our findings. For that we are very grateful. Please find below a point by point response to all comments (in yellow) and at the end we list all additional changes to the revised manuscript that were not mentioned in the responses.

Reviewer #1 (Remarks to the Author): with expertise in HLA-immunopeptidomics

Synopsis

The authors present an interesting study proposing HLA presented, RNA-derived variants as an immunologically relevant and potentially more prevalent source of neoepitopes than “classical” genomic somatic mutations. The authors leverage a proteogenomic workflow consisting of DNA- and RNA sequencing combined with HLA peptidomics across a set of 32 samples from 25 tumor types, resulting in the identification of 91 candidate neoantigens, the majority of which is derived from RNA variants. Immunogenicity testing of a subset of peptides and donors suggests memory T cell recognition of around 30% of these variant peptides.

Major Comments

While the study is appropriately sized and designed for an exploratory assessment of RNA-derived, HLA presented variants across tumor types, the central claims of tumor association/specificity of said variants requires further validation and their unambiguous molecular verification is mandatory. These two central prerequisites must be met prior to publication in Nature Communications. The authors use relaxed criteria for variant calling and mass spectrometric identification which is an appropriate way to improve sensitivity, as long as sequence verification of the variant peptides is performed. Due to the degeneracy of TCR recognition, the observed immunogenicity of some of the variant peptides alone cannot be deemed sufficient verification. Sequence confirmation by co-elution using spiked-in stable isotope labeled peptides as described in (Chong et al. 2020, <https://doi.org/10.1038/s41467-020-14968-9> and Fritsche et al., 2021, <https://doi.org/10.1016/j.mcpro.2021.100110>) should be performed to ultimately provide reliable frequency estimates for RNA- vs DNA derived variant peptides. This should be feasible in light of the limited number of variant peptide candidates and would help avoid inaccurate prevalence estimates as previously observed in the field of HLA peptidomics in the context of proteasomally spliced peptides (Lichti et al., 2022, DOI:10.1158/2326-6066.CIR-21-0727).

We would like to thank the reviewer for all the constructive feedback. We agree that peptide sequence verification is necessary for the identified candidates as quality control prior to further clinical translation and now performed this validation in our revised manuscript. Since we do not have any tumor material left (which is a common limitation for studies with primary tumor tissues), we could not perform “co-elution experiments using spiked-in stable isotope labeled peptides” as suggested by the reviewer and previously performed by Fritsche et. al. 2021 on

cell lines. In addition, although such an experiment may provide proof of principle in selected neoantigen candidates, this approach would likely be difficult to be implemented into a clinical workflow which was our translational goal. However, based on the reviewer's comments, we implemented additional quality controls for validation which may be helpful for selection of candidates for therapeutic applications. In particular, we synthesized the detected neoantigens, measured them with comparable LC-MS/MS settings and compared the acquired tandem-MS spectra from the synthetic peptides and the endogenous peptides. In addition, we performed the same comparison, using Prosit-predicted fragment ion intensities. We defined acceptance criteria and deemed endogenous neoantigen peptide candidates as verified if they either had a normalized spectral contrast angle (SA) (doi:10.1074/mcp.O113.036475) of 0.7 or greater with the synthetic or the Prosit-predicted spectrum. This SA cutoff of 0.7 has been previously established in other studies and was found to be appropriate for sequence verification (DOI: 10.15252/msb.20188503). After testing of 89 peptide neoantigen candidates, we found that 42 could be in fact verified using these criteria. A number of candidates were close to the SA cutoff and may still represent true hits which need to be further confirmed. Others with low SA values may not be safe enough to be followed for therapeutic targeting as they might actually not have been presented by the tumor. As the reviewer points out, our relaxed criteria are an appropriate way to improve sensitivity and are commonly used in the field of neoantigen identification due to the technical challenges and extremely low prevalence of these targets. This approach serves as a hypothesis generator for the identification of neoantigens and potential candidates are then validated extensively. Therefore, the candidates that could not be verified are likely a result of our relaxed criteria as the reviewer suspected. However, while stricter FDRs for peptide spectrum matching below 1% result in less peptides with a SA below 0.7, we still observed many peptides above the SA cutoff with q-values between 1 and 5% (in particular for pFind) that would have been missed if we would have used stricter criteria (shown in the new Suppl. Figure 9) rather than subsequent peptide verification. We also added the predicted retention time difference as a measure for peptide verification as to the reviewer's suggestion (described in the corresponding reviewer's comment below; new Suppl. Figure 10) and combined this information in the new main Figure 7.

We added an entire chapter in the revised manuscript that addresses this verification and the prevalence assessment of the candidate variants in normal tissue RNA-seq data that was also raised by the reviewer in another comment. We believe that these additional control analyses strengthen the reliability of our pipeline for the discovery of potential neoantigens and are very grateful to the reviewer for the constructive feedback.

*The claim of tumor association/specificity of the observed RNA variants cannot be supported by the provided data. Comprehensive assessment of **variant expression levels** across various, ideally matched, normal tissues are required to ascertain tumor-specificity and support the classification as RNA-derived neoantigens or to verify tumor-associated RNA-overediting and classify the peptides accordingly (see Zhang et al, 2018 for examples of RNA editing events in normal tissue, <https://doi.org/10.1038/s41467-018-06405-9>). This may be addressed by processing publicly available RNAseq repositories of normal tissues such as GTEx (www.gtexportal.org). Prematurely annotating variants as targets while not sufficiently screening normal tissues poses severe risks for on-target toxicities in later clinical use of such targets.*

Thank you for the comment and the helpful suggestion. To address this valid concern, we have now analyzed over 10,000 RNA-seq samples from 30 different normal tissues of the publicly available GTEx repository for the prevalence of neoantigen candidate variants. Out of 91 neoantigen candidates, 38 were completely absent in normal tissue RNA-seq samples in our extensive GTEx analysis and might present highly interesting neoantigen candidates based on this criterion. The other 53 candidates either showed high prevalence (n = 16 candidates; found in more than 5% of samples), intermediate prevalence (n = 6 candidates; found in 1 to 5% of samples), low prevalence (n = 12 candidates, found in 0.1 to less than 1% of samples), very low prevalence (n = 7 candidates; found in less than 0.1% of samples) in normal tissues or were defined as not available (N/A) based on expression data (n = 9 candidates; where the variant locus is expressed in less than 5% of normal tissue samples with at least 3 reads) (Figure 7, Suppl. Figure 11a). Of note, DNA-derived variants were not detected in normal tissues with our comprehensive analysis as expected.

In addition, we investigated tumor-associated RNA-overediting by comparing the variant frequency for the neoantigen candidates in tumor and normal tissues but only observed potential tumor-associated RNA-overediting in selected candidates, while the great majority did not show overediting (new Suppl. Figure 12).

As mentioned above, we added an entire chapter in the revised manuscript that addresses the prevalence of the candidate variants in normal tissue RNA-seq data from our GTEx analysis and discuss the importance of these quality control steps for our pipeline.

Minor Comments

- *DNA vs RNA variant calling (Figure 1, also in lines 539-558): Different callers were used for WES/WGS and RNAseq. Given that Strelka2 supports both DNaseq and RNAseq variant calling, it may have been beneficial to only use a single caller/pipeline to avoid calling bias which may affect the downstream classification of variants as DNA/RNA-only.*

The rationale why we used two different callers for WES/WGS and RNA-seq was that we observed superior calling properties for Mutect2 over Strelka2 for WES/WGS in mice (see publication <https://pubmed.ncbi.nlm.nih.gov/31907453/>). Therefore, we decided to use Mutect2 for WES/WGS to avoid oversight of potential somatic mutations.

- *The filtering strategy may have missed some known/annotated SNPs. For instance the proposed neoantigen ALSGHLETL is listed in dbSNP (rs17845226) and thus less likely a somatic mutation. To prove that the additional RNA-based events are bona-fide somatic mutations and not false positive calls requires an assessment of how reliably WES-confirmed mutations can be called by RNAseq. Even for WES high numbers of false positives for de novo calls are expected*

We thank the reviewer for raising this concern. For this reason, we checked all variants for SNPs with the dbSNP database. Population SNPs with a population allele frequency based on GnomAD over 1% and dbSNP over 5% were excluded. In addition, neoantigen candidates were manually re-evaluated for SNPs. In this specific example, the ID for

ALSGHLETL (Chr:Pos 9:33624565) is incorrectly matched to rs17845226. The mentioned SNP with the ID rs17845226 is on Chr15 while ALSGHLETL (Chr:Pos 9:33624565) is on Chr9. Moreover, we screened our candidates with three different peptide databases (PeptideAtlas, PepBank, IEDB) for already known (immunogenic) peptides.

Although, RNA-seq could identify the majority of WES/WGS-derived variants (Suppl. Fig. 4a; about 60%), it could not identify all. This is why both datasets as well as the control DNA obtained from the patients' blood are crucial to reduce false positives. Our analysis shows that the majority of these additional RNA-based variants have a canonical sequence at the variant locus in the matched WES/WGS data (Figure 3b, "RNA only" variants), indicating that they are not derived from somatic mutations. To assess if bona fide somatic mutations are present in the "RNA only" variants that were not covered by WES/WGS was not feasible here. In our data the majority of DNA-based events were derived from WES. Therefore, noncoding regions were insufficiently covered. To overcome this in future studies, performing WGS on all samples could offer some insights here.

To clarify which neoantigen candidates have no WES/WGS coverage (and potentially could be derived from missed somatic mutations) we added this information for each candidate in a new Figure 7 together with the quality control information described above.

- *Summary statistics of DNA/RNA variants vs canonical reads would be highly informative to compare coverage, minor allele fractions, number of reads, normalized expression*

We provide the summary statistics DNA/RNA variants vs canonical reads for the number of reads here:

Summary read count of all variants vs canonicals

Summary read count of neoantigen candidates variants vs canonicals

	Median of reads for all variants	Median of reads for the 91 neoantigen variants
Blood variants (DNA)	0	0
Blood canonicals (DNA)	99	83
Tumor variants (DNA)	3	4
Tumor canonicals (DNA)	117	99
Tumor variants (RNA)	5	10
Tumor canonicals (RNA)	15	15

Since tumor samples are never pure but contain a lot of tumor stroma, it is to be expected that the tumor canonical read count is higher than the variant read count. Moreover, the expected high level of heterogeneity between different tumor samples can also play a role here.

Please note that the individual values for each variant can be found in the Suppl. Tables.

- *The majority of peptides listed in Supplementary Table S3 are misleadingly called tumor-associated peptides, yet the only evidence is the assignment in Protein atlas as “cancer-gene” based on COSMIC mutations in the protein of origin and annotation of that protein as plasma biomarker in cancer. No evidence for over-expression in RNA or on peptide-level in cancer tissues is provided. Furthermore, 14 of 18 of these peptides are found in the HLA Ligand Atlas (<https://hla-ligand-atlas.org>) thus confirming presentation of these peptides on healthy tissues.*

That is correct and we agree with this assessment. These peptides are only from cancer-associated genes and evidence of their tumor association will need to be investigated in future studies. Therefore, we removed the Suppl. Figure 5b and the corresponding Suppl. Table 3 since highlighting specific targets here would need further validation which is out of scope and would deviate from the focus of our neoantigen discovery pipeline in this study.

- *HLA peptidomics: FDRs for identification of variant peptides are very relaxed thus clearly calling for peptide verification prior to downstream immunogenicity testing. Variant peptides from two different search pipelines (pFind @5% FDR & pFind no FDR + Prosit/Percolator spectral rescoreing @5% FDR) were utilized to derive a combined list of putative variant peptides. Apart from global FDR potentially being inaccurate for low prior probability subsets of peptides, the combination of two sets of search results for the same raw data further inflates potential false positive IDs. An indication of potential inflation in the non-canonical IDs is given by the low number of HLA binders in table S4 (9 of 23) compared to S3.*

As described above, we agree with the need for peptide verification and performed it by two means: synthetic peptides and prediction of peptides by established deep learning models (<https://doi.org/10.1038/s41467-021-23713-9>) (new Figure 7). In addition, we added the predicted retention time difference as a measure for peptide verification as to the reviewer’s suggestion (described in the corresponding reviewer’s comment below; new Suppl. Figure 10).

We agree with the statement that merging search results can result in inflated identifications. Typically, this is most prevalently observed on protein level, not so much on PSM or peptide level. Within the class of low abundant HLA peptides, patient-specific mutations are even harder to identify which makes the use for relaxed criteria necessary. Moreover, de novo assembled personalized patient databases as we use here provide additional specificity by limiting the identity and number of non-canonical peptides that were probed into the LC-MS/MS data.

We agree that the global FDR might be slightly inflated, however we consciously selected a less stringent FDR cutoff of 5% because biochemical evaluation of potential candidates was performed downstream of LC-MS/MS detection. By adding peptide verification and

prevalence assessment of the neoantigen candidate variants in normal tissues (as suggested by the reviewer and described above), we observed an enrichment of predicted binders and candidates that were identified with both search tools in the group of validated neoantigen candidates which we discuss in the revised manuscript. This supports our validation approach for our pipeline. Moreover, many candidates that passed peptide verification would have been missed if we would have used stricter FDRs below 1% (new Suppl. Figure 9) as discussed above.

- *Compare prediction scores, spectra and HLA alleles to be predicted to bind neoantigens*

We performed peptide verification in the revised manuscript. Here, we compared the experimental retention time and the predicted retention time as well as the spectral angle of the Prosit-predicted or synthetic peptide with the 91 neoantigen candidates. This can be found in the new chapter and is summarized in Figure 7.

The predicted HLA alleles for the binding of the neoantigen candidates with MHCFlurry and netMHC can be found in Suppl. Table 4. Moreover, we discuss in the revised text that we observed an enrichment of predicted binders in the group of validated neoantigen candidates.

- *The authors have previously presented predicted retention time as an important parameter apart from spectral correlation/angle for improving specificity and sensitivity of peptide rescoring pipelines (<https://www.msaid.de/inferys>). Retention time was not used as a scoring parameter in this study and should be included, at least for post hoc assessment of variant peptide IDs, e.g. as performed in Chong et al. 2020, <https://doi.org/10.1038/s41467-020-14968-9> by correlation of predicted vs experimental retention times*

Thank you for the suggestion. We added the comparison of experimental retention time (RT) vs predicted RT as an additional scoring parameter for peptide verification. This can be found in the new chapter and is summarized in Figure 7. We also provide the dot plot of experimental vs predicted RTs as well as the distribution of the errors for all peptides in the new Suppl. Figure 10.

Mismatch was assessed as outliers according to the extreme of the upper whisker of the absolute error boxplot for all datapoints (± 8.56 min). Of all test neoantigen candidates ($n = 89$), 46 matched and 26 mismatched. For some peptides (between the experimental RT range of 9 to 17 min) no accurate RT error could be predicted according to the distribution of canonical peptides and these were considered deviations ($n = 17$).

- *Summary statistics such as FDR, spectral correlation, %predicted binders, delta predicted RT should be provided to compare variant vs canonical peptide IDs*

Please find summary statistics for the q-values of pFind and Prosit in the graph below. The percentage of the predicted binders for the neoantigen candidates are now discussed in the new validation chapter. The individual binding prediction scores for each candidate can be found in Suppl. Table 4. The predicted vs experimental RT and error distribution can be found in the new Suppl. Figure 10.

Summary q-values of all neo epitopes vs canonical peptides

- *Relevant information for reproduction of peptide identifications are missing (e.g. mass tolerance used for MS1 and MS2, discrimination between static and dynamic PTMs, considered charge states, ...). MS2 spectra of the proposed variant peptides are not provided.*

The additional information was added to the method section of the revised manuscript. The MS2 spectra vs the corresponding synthetic or Prosit-predicted spectra will be provided as additional data in the supplementary material.

- *Fig. S5A: "Peptides per gram tumor" should not be plotted as shown here as it suggests linear correlation of tumor input and peptides present / identified, which is not (necessarily) the case. Extrapolation is misleading.*

We removed S5A from the figure.

Reviewer #2 (Remarks to the Author): with expertise in proteogenomics, RNA editing

The paper by Tretter et al. is focused on tumor neoantigen identification on the level of HLA-bound peptides in 32 patients partly already characterized in a specific MASTER program. The novelty was that authors considered specifically neoantigens originated from RNA variants. They were shown to be less variable than mutation-associated genomic mutations and, in further perspective, are considered as components for cancer vaccines.

The results compose an important multiomic dataset for further use. I would not say that they may have immediate clinical importance, as neoantigen vaccines still persist on the stage of I phase clinical trials and may never leave them for a wider practice (which authors admit in the discussion). At the same time, the new data presented have substantial basic significance. I am, thus, generally positive about this paper.

We thank the reviewer for the positive assessment of our study and for pointing out the value of our comprehensive multiomic dataset.

It is impossible to check all the huge data array presented here for their quality during the review. Let's just trust the author's reputation in this respect. Which could be done for improvement of the manuscript, it is to clarify some details of study in the text and to explain some decisions in data processing. Below I present some specific comments for authors' consideration. Most of them relate to the peptidomic part as nucleic acid pipelines and methods are better established and standardized.

1. *DNA vs. RNA neoantigens and nucleic acid polymorphism. The important result which, in spite it was expected, was that DNA variants are more variable, i.e. personalized than RNA variants. Please discuss it more elsewhere why we observe this result. In the present form, the discussion is focused on future clinical applications that could not happen. Less attention is paid to the basic cancer biology.*

Thank you for this feedback. We discuss this now in the revised manuscript and included additional text regarding the basic cancer biology in the discussion section. However, we did not go too deep in the description of underlying mechanisms or the biological relevance of these neoantigen candidates because this would be out of scope of our current study focusing mainly on identification and validation of neoantigen candidates including RNA variants. Promising candidates will need to be elucidated in future mechanistic studies. Here, we want to provide a multiomic pipeline for neoantigen discovery and raise awareness of the potential importance of RNA-derived neoantigens.

2. *In relation to the previous comment, please discuss more the biology of RNA variants, including those which compose neoantigens. The majority of them are A-to-G and are mentioned in Rediportal which means they are, most likely, products of ADAR RNA editing. How could occur the rest of them? This was not properly discussed.*

We agree and discuss that now more in the discussion section of the revised manuscript.

3. *Not all RNA neoantigens elicit immune response in ELISPOTA tests. We could say even that many of them failed to do so. Maybe, some of them are not NEO antigens. As A-to-I RNA editing may happen before thymus development (and a formation of the repertoire of immunological tolerance), some of these antigens formed via RNA editing could be presented to the immune system. Maybe, this could be mentioned in Results or Discussion sections.*

That is absolutely correct and could have multiple possible reasons:

1. T cell dysfunction of autologous T cells from heavily treated patients
→ potential failure to detect an immunogenic neoantigen

2. Low frequencies of neoantigen-specific T cells in the blood due to immune editing events as deletion of overstimulated T cells
→ potential failure to detect an immunogenic neoantigen
3. Thymic tolerance of RNA-editing derived antigens
→ potentially no neoantigen but derived from physiological RNA-editing
4. Missing coverage of canonical DNA sequences in non-coding regions with WES for some RNA variants
→ potentially missed somatic mutations and not derived from RNA editing

(1) and (2) may represent scenarios for true neoantigen candidates that did not elicit an immune response. These could still be interesting immunogenic targets.

With respect to (3), we analyzed for the revision over 10,000 RNA-seq samples from 30 different normal tissues of the publicly available GTEx repository for the prevalence of identified neoantigen candidates. 38 of the 91 neoantigen candidates were not present in normal tissues but a portion of the candidates were found frequently in healthy samples (n = 16 candidates; found in more than 5% of samples) and are likely no neoantigens (Figure 7, Suppl. Figure 11a).

As to another reviewer's helpful comments, we also performed peptide sequence verification for the identified candidates and performed these in our revised manuscript since the TCR repertoire is capable of eliciting immune responses against random foreign peptides. Here, we analyzed the synthetic peptides and the Prosit-predicted peptides and matched them to the endogenous neoantigen peptide candidates (described in detail in the revised text). We also added the retention times as a measure for peptide verification as to this reviewer's suggestion and combined this information in the new main Figure 7. In Figure 7 we also added additional information of the neoantigen candidates that are often missed in extensive supplementary tables, like the coverage of canonical DNA sequences with WES (4).

We added an entire chapter in the revised manuscript that addresses this verification and the prevalence of the candidates in normal tissue RNA-seq data. There we also discuss the points regarding the ELISpot data that was raised by the reviewer. We believe that these additional control analyses strengthen the reliability of our pipeline for the discovery of potential neoantigens.

4. *Identification of variant-containing peptides (HLA ligands). This stage is technically problematic because the presence in the preparation of non-relevant peptides with low affinity to HLA adds difficulty as well as false assignments of PSMs with are not constrained by trypsin cleavage. Some questions arose to this parts that would be better replied in the revised paper.*

4.1 Why so called wild-type, consensus genome-encoded peptides were analyzed (and results described) separately from DNA and RNA mutated peptides? Theoretically, if we take a customized (personalized) database for each sample which contains consensus and personalized sequences, we may get right PSMs for both "wild-type" and variant peptides.

We have decided to analyse the peptidome data set in two separate analysis as different FDRs of 1 % and 5% have been used for the separate questions. For the analysis of the canonical peptide repertoire, 1% FDR was applied as here no further validation step (cross-validation with the genomic and transcriptomic data) was added and thus more stringent calling was deemed necessary. Furthermore, all patient samples were analysed in one big search together for the canonical peptide repertoire. In contrast, for neoantigen identification, patient samples had to be analysed in separate searches as for each tumor sample a specific reference data set had to be used. Moreover, the FDR was set to 5% to improve the sensitivity for neoantigen identification. In the revised manuscript we show that less strict FDRs with subsequent validation improve sensitivity, since many candidates that passed peptide verification would have been missed if we would have used stricter FDRs below 1% (new Suppl. Figure 9) as discussed in the new validation chapter.

4.2. *I could not find in results a direct indication of how many "wild-type" peptides of identified did not dock eventually to HLA-I of corresponding alleles for each sample. This number is important for the perception of the results. (I could miss something in Supplementary Files, if so, I am sorry for this).*

We have exemplified this information for 4 patients in Suppl. Figure 7 but we agree with the reviewer that this information is important and added a new Suppl. Table 3 in the revised manuscript with this information for all patients.

4.3. *If I understood correctly, for variant HLA-I-bound peptide search, all variants which could potentially be coding (even outside recognized human ORFs) were read by 3 potential reading frames, admixed to consensus genomic database and then searched. An additional measure which made the output more filtered was the prediction with Prosit. The question is, how many records were added to the search database via this 3-frame reading. How much do they outnumber a conventional database? It was shown, for example, that addition of whole COSMIC (all cancer mutations known) to the consensus database led to the so called database inflation and dramatically increased a real FDR in spite of 1% FDR conservation by the software (in the target-decoy method). At least, deduced 3-frame variants must be filtered separately from genomic peptides which are much more likely to be identified (so called group-specific FDR), which was not done.*

Thank for raising this point. The number of new entries (patient specific database) is actually smaller than the conventional consensus genomic database:

conventional: total = 418,865 entries

new entries: mean = 7,350, range = 40-20,000, total = 308,720 entries

This is because not every possible ORF is selected, but only valid ones were added. That means that each potential ORF is checked for a valid start and stop codon paired within the same reading frame. In our case, we did not add all known cancer mutations, but just the patient-specific ones. Moreover, additional filtering was done with Mutect2 regarding quality to reduce the number of false positives.

5. *Data availability. A significant feature of the paper is the high-quality multiomic data generated. Now these data are available from the site of one of the institutes involved in the study. It would be preferable to make them available from one of public repositories. Unfortunately, data on institute's sites tend to stay neglected and disappear after finishing someone's postdoc and PhD contracts.*

We absolutely agree with the reviewer and all multiomic data have been deposited to public repositories. The dataset accession numbers are provided in the data availability statement in the revised manuscript.

Minor comment:

In the text, a genetic variant as a term was applied to both DNA and RNA variants. I do not think this is exact, as RNA variant is not genetic, but phenotypic. I think it may be modified to avoid confusion. Further, 'wild-type' definition sounds weird in respect of the human data (yes I also used it, but it is really argotic expression, in that case, not scientifically exact.

Thank you for this feedback. We agree and revised the text to not refer to both as genetic. In addition, we changed wild-type to canonical.

Reviewer #3 (Remarks to the Author): with expertise in HLA-immunopeptidomics, Ag presentation

The authors found that considerable numbers of genetic variants were generated at the RNA level (e.g. by A-to-I RNA editing) across a variety of tumor types. Mutated peptides were translated and some were displayed by HLA class I molecules. Importantly, patient CD8+ T cells recognized those mutated peptides, suggesting that RNA variants, which are likely intact at the DNA level, served as a novel source of antigens potentially leading to anti-tumor immune responses. In addition, variant calling using RNA enabled to detect non-coding RNA-derived neoantigen candidates. These findings are interesting. To my knowledge, it is still difficult to identify real neoantigens that induced spontaneous patient T cell responses in vivo. This may be due to lack of methodology or the presence of yet unexplored sources of neoantigens. Therefore, the findings in this study are of importance and contribute to the research field of tumor antigens and T cell surveillance. Here, I have two questions.

We would like to thank the reviewer for the positive assessment of our study.

1. Even if RNA variants gave rise to immunogenic antigens and could induce reactive T cell responses, it is still unclear if target antigens are tumor-specific. In contrast to somatic DNA mutation, RNA editing can occur and possibly lead to HLA presentation of same immunogenic antigens in normal cells. In this case, antigens may serve as autoantigens. The authors demonstrated antigen recognition by TILs in Fig 6A, which suggested the tumor specificity of

this particular peptide; however, most identified antigens were recognized only by autologous PBMCs across patients. Do authors have additional evidence showing their tumor specificity?

That is a very important point that was also raised by another reviewer. In order to elucidate this, we performed additional analyses. We analyzed over 10,000 RNA-seq samples from 30 different normal tissues of the publicly available GTEx repository for the prevalence of neoantigen candidates. Out of 91 neoantigen candidates, 38 were completely absent in normal tissue RNA-seq samples in our extensive GTEx analysis and might present potential neoantigens. The other 53 candidates either showed high prevalence (n = 16 candidates; found in more than 5% of samples), intermediate prevalence (n = 6 candidates; found in 1 to 5% of samples), low prevalence (n = 12 candidates, found in 0.1 to less than 1% of samples), very low prevalence (n = 7 candidates; found in less than 0.1% of samples) in normal tissues or were defined as not available (N/A) based on expression data (n = 9 candidates; where the variant locus is expressed in less than 5% of normal tissue samples with at least 3 reads) (Figure 7, Suppl. Figure 11a).

In addition, we investigated tumor-associated RNA-overediting by comparing the variant frequency for the neoantigen candidates in tumor and normal tissues but only observed potential tumor-associated RNA-overediting in selected candidates, while the great majority did not show overediting (new Suppl. Figure 12).

We added an entire chapter in the revised manuscript that addresses the expression of the candidates in normal tissue RNA-seq data from our GTEx analysis and discuss the importance of these quality control steps for our pipeline.

2. A single tumor mass should be heterogeneous regarding an RNA WT/variant expression ratio. Is it possible to estimate clonality of each RNA variant within a tumor and clarify if the clonality is associated with any biological consequences (e.g. HLA presentation or induction of T cell responses)?

Indeed, this is a very interesting question. We provide the variant frequency for each neoantigen candidate in Suppl. Table 4, but clonality for the RNA variants in our approach cannot be investigated because we did bulk RNA-seq of tumor samples. To address this, we would need to perform single cell RNA-seq to acquire the expression level for each individual tumor cell what was out of scope for the size of our cohort at the time. In addition, we investigated tumor-associated RNA-overediting by comparing the variant frequency for the neoantigen candidates in tumor and normal tissues in the revised manuscript. We only observed potential tumor-associated RNA-overediting in selected candidates, while the great majority did not show overediting (new Suppl. Figure 12). We agree with the reviewer that we would expect a heterogeneous expression ratio in a single tumor mass but this would need to be elucidated in future studies.

Changes to the text of the revised manuscript are marked in yellow. In addition, please find below a checklist of items that were removed from the revised manuscript during revision that were not already mentioned in the responses above:

- We shortened the title by removing "...correlating with T-cell infiltration". Although we still observed these correlations in patients with immunogenic validated neoantigen candidates (new Suppl. Figure 14), the sample size is low. Therefore, we removed this emphasis from the title. In our opinion the major finding is that RNA is an important source for neoantigen candidates, which remains unchanged in the revised manuscript.
- In Suppl. Table 4 we removed the column "Reads PBMC RNA Ref:Alt" since it could be misleading. Here, we used PBMC DNA to assess if reads of RNA variants were found at the DNA level in PBMCs.
- By carefully re-evaluating every ELISpot assay during the revision we observed that one assay's negative control could have been compromised (low cell count) and removed this assay for that reason. This results in the removal of 3 immunogenic neoantigen candidates from patient ImmuneNEO_4, reducing the count of detected immunogenic candidates from 24 to 21.
- We removed the annotation for variants that were reported in the databank REDportal as A-to-I RNA editing events in the figures with nucleotide exchange pattern because these were reported in normal tissue expression data and not cancer.
- We re-evaluated all 91 neoantigen candidates for the publication of newly discovered SNPs during the revision. This is a process that we also performed manually before submission to identify SNPs that were only published in dbSNP after variant calling was performed. We identified a SNP (rs1213038645) that was just recently released in 21st of September 2022 (around the time of submission of our manuscript) for our neoantigen candidate 3B. Therefore, we removed the candidate in the revised manuscript, reducing the number of neoantigen candidates from 91 to 90. Off note, this candidate did not pass our validation process in the revised manuscript because of high prevalence in GTEx samples (above 5% of all samples), as expected from a common SNP. This further supports the power of our validation approach.

REVIEWERS' COMMENTS

Reviewer #1 (Remarks to the Author):

The authors have diligently addressed the concerns raised during the initial review and have significantly strengthened the manuscript by performing extensive additional evaluation of variant peptide identity as well as their prevalence in normal tissues on the RNA-level. The authors provide a breakdown of the associated attrition of candidate variants in a new chapter and main figure, thereby giving valuable insights into attrition factors in (neo-) antigen identification using proteogenomic workflows and the pitfalls of immunogenicity testing for epitope verification. The revised manuscript is recommended for acceptance in Nature Communications.

Reviewer #2 (Remarks to the Author):

No further comments. All my remarks were adequately considered and met when it was needed.

Reviewer #3 (Remarks to the Author):

The authors have satisfactorily addressed the concerns raised by this reviewer.

Point-by-point response to the reviewers' comments

We thank all reviewers for the fast processing time and their positive assessment of our revised manuscript. Moreover, we would like to express our gratitude to the reviewers for the very constructive and professional peer review. Please find below a point-by-point response to all comments (in yellow):

Reviewer #1 (Remarks to the Author):

The authors have diligently addressed the concerns raised during the initial review and have significantly strengthened the manuscript by performing extensive additional evaluation of variant peptide identity as well as their prevalence in normal tissues on the RNA-level. The authors provide a breakdown of the associated attrition of candidate variants in a new chapter and main figure, thereby giving valuable insights into attrition factors in (neo-) antigen identification using proteogenomic workflows and the pitfalls of immunogenicity testing for epitope verification. The revised manuscript is recommended for acceptance in Nature Communications.

Thank you for the constructive feedback during the initial review and for your kind words regarding our revised manuscript.

Reviewer #2 (Remarks to the Author):

No further comments. All my remarks were adequately considered and met when it was needed.

Thank you for all the helpful remarks during the revision and for your endorsement of our revised manuscript.

Reviewer #3 (Remarks to the Author):

The authors have satisfactorily addressed the concerns raised by this reviewer.

Thank you for your initial review and the positive evaluation of our revised manuscript.